# High-efficiency and stable short-delayed fluorescence emitters with hybrid long- and short-range charge-transfer excitations

Guoyun Meng[1], Hengyi Dai[1], Qi Wang[1], Jianping Zhou[1], Tianjiao Fan[1], Xuan Zeng[1], Xiang Wang[1], Yuewei Zhang [1,2], Dezhi Yang[3], Dongge Ma [3], Dongdong Zhang [1,2] ✉ & Lian Duan [1,2] ✉

The pursuit of ideal short-delayed thermally activated delayed fluorescence (TADF) emitters is hampered by the mutual exclusion of a small singlet-triplet energy gap ($\Delta E_{ST}$) and a large oscillator strength ($f$). Here, by attaching an multiresonance-acceptor onto a sterically-uncrowded donor, we report TADF emitters bearing hybrid electronic excitations with a main donor-to-acceptor long-range (LR) and an auxiliary bridge-phenyl short-range (SR) charge-transfer characters, balancing a small $\Delta E_{ST}$ and a large $f$. Moreover, the incorporation of dual equivalent multiresonance-acceptors is found to double the $f$ value without affecting the $\Delta E_{ST}$. A large radiative decay rate over an order of magnitude higher than the intersystem crossing (ISC) rate, and a decent reverse ISC rate of $>10^6 \, s^{-1}$ are simultaneously obtained in one emitter, leading to a short delayed-lifetime of ~0.88 μs. The corresponding organic light-emitting diode exhibits a record-high maximum external quantum efficiency of 40.4% with alleviated efficiency roll-off and extended lifetime.

The ability to harvest the otherwise dark triplet excitons for emission is essential to improve the efficiency of organic light-emitting diodes (OLEDs) under electrical excitation[1–3]. Following the success of orga-nometallic complexes containing precious metals, emitters with ther-mally activated delayed fluorescence (TADF) have come to the forefront in recent years as their notable merits of 100% exciton utili-zation efficiency using noble-metal-free organic molecules[4–8]. The distinct character of a TADF emitter is its energetically close lowest-energy singlet ($S_1$) and triplet ($T_1$) excited states, enabling the up-conversion of dark $T_1$ excitons into radiative $S_1$ excitons via reverse intersystem crossing (RISC) at room temperature[9]. For an ideal TADF emitter, a fast exciton decay process is essential, which is governed by fast triplet up-conversion and singlet radiation processes simulta-neously. The former requires a small singlet-triplet energy gap ($\Delta E_{ST}$) while the latter a large oscillator strength ($f$), which, however, are conflicting factors as they showed contrary dependence on the orbital

overlap integral[10]. For most TADF emitters, long-delayed lifetimes ($\tau_D s$) of ~μs-ms are thereof observed, leading to severe bimolecular exciton annihilations under high current density[11]. Conceptually advancing molecular design strategy to break the trade-off between a small $\Delta E_{ST}$ and a large $f$ therefore remains ongoing pursuits in literature.

To date, the traditional design principle for most TADF emitters can be simplified as donor (D)-acceptor (A) or D-π-A architectures with one or multiple donors or acceptors, of which the TADF behaviors are highly affected by the molecule electronic excitation characters. For TADF molecules with completely separated frontier molecular orbital distributions, a solely donor-to-acceptor long-range charge-transfer (LR-CT) excitation can be expected, which favors a small $\Delta E_{ST}$ but also a small $f$. Conventional wisdom to solve this issue is to introduce some orbital overlap on the π-bridge to hybridize localized excitation (LE) to enlarge $f$ value in addition to the main LR-CT character. Nevertheless, discreetly balancing a proper portion of LR-CT and LE excitations to

[1]Key Laboratory of Organic Optoelectronics, Department of Chemistry, Tsinghua University, Beijing 100084, P. R. China. [2]Laboratory of Flexible Elec-tronics Technology, Tsinghua University, Beijing 100084, P. R. China. [3]Institute of Polymer Optoelectronic Materials and Devices State Key Laboratory of Luminescent Materials and Devices, South China University of Technology, Guangzhou 510640, P. R. China. ✉e-mail: ddzhang@mail.tsinghua.edu.cn; duanl@mail.tsinghua.edu.cn

make a balance of a small $\Delta E_{ST}$ and a large $f$ is rather difficult in molecular design and most emitters still face the inevitably enlarged $\Delta E_{ST}$s with only few emitters can reach a compromise[12,13]. Besides, an even tougher but often ignored situation is the counter-effect from the intersystem crossing (ISC) process, which can occur much more rapidly than radiative decay and thus would increase the singlet-to-triplet spin-flip transition cycles to prolong the $\tau_{D}$s of TADF emitters[14–16]. Therefore, though some efficient TADF emitters have been reported, most molecules with solely LR-CT or hybrid LR-CT/LE excitations still suffer from long-delayed components induced significant efficiency roll-off and poor operation stability.

Recently, multiresonance (MR) molecules have also emerged as a new family of TADF emitters by reducing $\Delta E_{ST}$s through the unique offset of the frontier molecular orbital distribution on single atoms[17–20]. The resulting short-range (SR) CT excitation endows MR molecules with the merits of localized states with large $f$ values for extremely fast radiative decay rates ($k_{r}$s), even larger than their ISC rates ($k_{ISC}$s). Nonetheless, only moderate $\Delta E_{ST}$s can be obtained by MR emitters and according to the Marcus-Levich-Jortner theory, the small Huang-Rhys factors of those molecules also render them rather slow RISC rates ($k_{RISC}$s) in the range of $10^{4}$ s$^{-1}$ and even milli-second-scale $\tau_{D}$[21]. Inspired by the LR-CT/LE type emitters, we here envisioned that hybridizing an auxiliary SR-CT with the main LR-CT excitation may not only avoid their disadvantages but also combine their complementary advantages to afford a new paradigm of TADF molecules simultaneously with a small $\Delta E_{ST}$ for a fast RISC process and a large $f$ for an efficient radiative decay.

With this in mind, we demonstrated here a strategic implementation of TADF emitters with hybrid LR/SR-CT excitations by incorporating MR acceptor groups onto a sterically uncrowded donor segment (Fig. 1a). This architecture, on one hand, created a hybrid electronic excitation with a main donor-to-acceptor LR-CT and an auxiliary bridge-phenyl SR-CT characteristics, balancing a small $\Delta E_{ST}$ and a large $f$. On the other hand, the adoption of equivalent dual MR acceptors was unveiled to double $f$ without influencing $\Delta E_{ST}$ and also enlarged the horizontal dipole ratio for enhanced light extraction. A large $k_{r}$ of $6.0 \times 10^{7}$ s$^{-1}$, nearly 20 times higher than $k_{ISC}$, and a decent $k_{RISC}$ of $1.2 \times 10^{6}$ s$^{-1}$ were thereafter accomplished concurrently, establishing an ideal exciton dynamic model of $k_{r} \gg k_{ISC} \sim k_{RISC} > 10^{6}$ s$^{-1}$ for a short sub-microsecond-scale $\tau_{D}$ of 0.88 μs. The corresponding device exhibited an unprecedentedly high maximum external quantum efficiency (EQE$_{max}$) of 40.4% with alleviated efficiency roll-off and prolonged device stability. To narrow the emission spectra, those molecules were further adopted as sensitizers for an MR emitter, achieving an EQE$_{max}$ of 38.4% with a full-width at half-maximum (FWHM) of 25 nm. Our achievement here shatters the stereotypical physicochemical views of TADF emitters limited by the mutual exclusion of a small $\Delta E_{ST}$ and a large $f$, potentially revolutionizing the molecular design principle and unlocking the full potential of TADF OLEDs.

## Results and discussion
### Molecular synthesis and computational results
The structure of the proof-of-the-concept molecule, 5-(3,11-bis(tri-fluoromethyl)−5,9-dioxa-13b-boranaphtho[3,2,1-de]anthracene-7-yl)−11-phenyl-5,11-dihydroindolo[3,2-b]carbazole (1BOICz), was provided in Fig. 1b, constructed by attaching a trifluoromethyl (CF$_3$) group substituted oxygen-bridged boron (CF$_3$-BO) group on to a 5-phenyl-5,11-dihydroindolo[3,2-b]carbazole (32bICz) segment. The BO derivatives have been proven to possess obvious MR properties[17,22] and thus were adopted as MR acceptor units. While 32bICz was a well-known donor group. For comparison, we also selected a reference TADF emitter, 5-(4-(4,6-diphenyl-1,3,5-triazin-2-yl)phenyl)−11-phenyl-5,11-dihydroindolo[3,2-b]carbazole (1TICz)[23]. We first performed the time-dependent DFT (TD-DFT) calculations to analyze the distributions of

the highest occupied molecular orbital (HOMO) and the lowest unoccupied molecular orbital (LUMO) and excitation characters of those molecules (Supplementary Table 1). Derived from the sterically uncrowded structures of both donor and acceptor planes, only moderate dihedral angles ($\theta$) in the range of 46-52° were observed for both compounds (Fig. 1b). As a result, in addition to the mainly separated frontier molecular orbitals (FMOs) on the donor and acceptor groups, the moderate twist motifs render obvious both HOMO and LUMO residence on their phenylene bridge but in rather different behaviors (Fig. 1c). For 1TICz, a clear localized π-bonding orbital distribution was observed on the phenylene bridge with significant HOMO-LUMO overlap, thus creating a hybrid orbital distribution combining LR-CT and LE characteristics. Therefore, only a moderate $\Delta E_{ST}$ of 0.193 eV together with a rather high $f$ value of 0.2361 was obtained for 1TICz. In terms of 1BOICz, its FMO distribution on the phenylene bridge showed a clear MR behavior with HOMO on the attached carbon atom and the carbon atoms positioned *meta* to it while the LUMO on the other carbon atoms. Such alternative electron-rich and electron-deficient regions on single atoms would thereof form the SR-CT transition for 1BOICz[22]. The natural transition orbital (NTO) analysis of S$_1$ and T$_1$ for both molecules was also conducted as illustrated in Supplementary Fig. 2. Clearly, in agreement with its SR-CT character on the bridge-phenyl ring, 1BOICz possessed a relatively larger ratio of CT transition than 1TICz. Interestingly, though its main CT transition, 1BOICz still exhibited a decent $f$ value of 0.1428, much larger than most donor-acceptor type TADF emitters, particularly those with twisted structures[24]. The reason should be attributed to the SR-CT transition on the bridge-phenyl ring, which has proved to favor a large $f$. More intriguingly, a small $\Delta E_{ST}$ of 0.1054 was also observed given the small orbital overlap. Therefore, the hybrid LR/SR-CT orbital distribution characters of 1BOICz render it more advantageous than the LR-CT/LE type of 1TICz in balancing a small $\Delta E_{ST}$ and a large $f$ value.

To compensate the sacrifice of $f$ value for 1BOICz, 5,11-bis(3,11-bis(trifluoromethyl)−5,9-dioxa-13b-boranaphtho[3,2,1-de]anthracen-7-yl)−5,11-dihydroindolo[3,2-b]carbazole (2BOICz) was further developed. The FMOs of 2BOICz inherit the hybrid LR/SR-CT orbital distribution characters of 1BOICz, except for the equivalent LUMO distributions on dual acceptors. Noting that the theoretical results revealed a similar S$_1$ and $\Delta E_{ST}$ for 2BOICz compared with that of 1BOICz, but a doubled $f$ value of 0.3498 (Fig. 1c). The plausible reason should be assigned to the dual equivalent MR acceptors of 2BOICz, which can create dual equivalent emitting channels to double the $f$ value. Considering that the $f$ value of 2BOICz is even higher than 1TICz, this means the decreased FMO overlap by SR-CT distribution can be compensated by the equivalent acceptors without sacrificing a small $\Delta E_{ST}$. In our previous works, TADF emitters with multiple but not equivalent donors and/or acceptors have been developed and thus no equivalent multiple emitting channels are being formed[25]. The high $f$ value of 2BOICz is even comparable with or larger than that of the conventional fluorophors with locally excited (LE) states and should lead to both fast radiative decay and RISC process combining with its small $\Delta E_{ST}$, which is exactly the aim of this molecular design.

To further clarify the superiority of the LR/SR-CT type orbital distributions, we simulated the corresponding $\Delta E_{ST}$ and $f$ values by varying the $\theta$ between the donor and acceptor of the three compounds (Fig. 1d–f), of which the orbital distribution changes were also provided in Supplementary Fig. 3. For 1TICz, with increased $\theta$ value, the excitation type was changed from LR-CT/LE to solely LR-CT and thus greatly varied both $\Delta E_{ST}$ and $f$ values. A small $\theta$, that is corresponding to a hybrid LR-CT/LE excitation, would sharply increase the $\Delta E_{ST}$ value while a large $f$. While a large $\theta$, that is corresponding to a solely LR-CT excitation, would greatly decrease the $f$ value while a small $\Delta E_{ST}$. Balancing a proper portion of LR-CT and LE excitations to compromise a small $\Delta E_{ST}$ and a large $f$ faces a formidable challenge in molecular design. Intriguingly, in terms of 1BOICz, a small $\theta$ greatly enhances

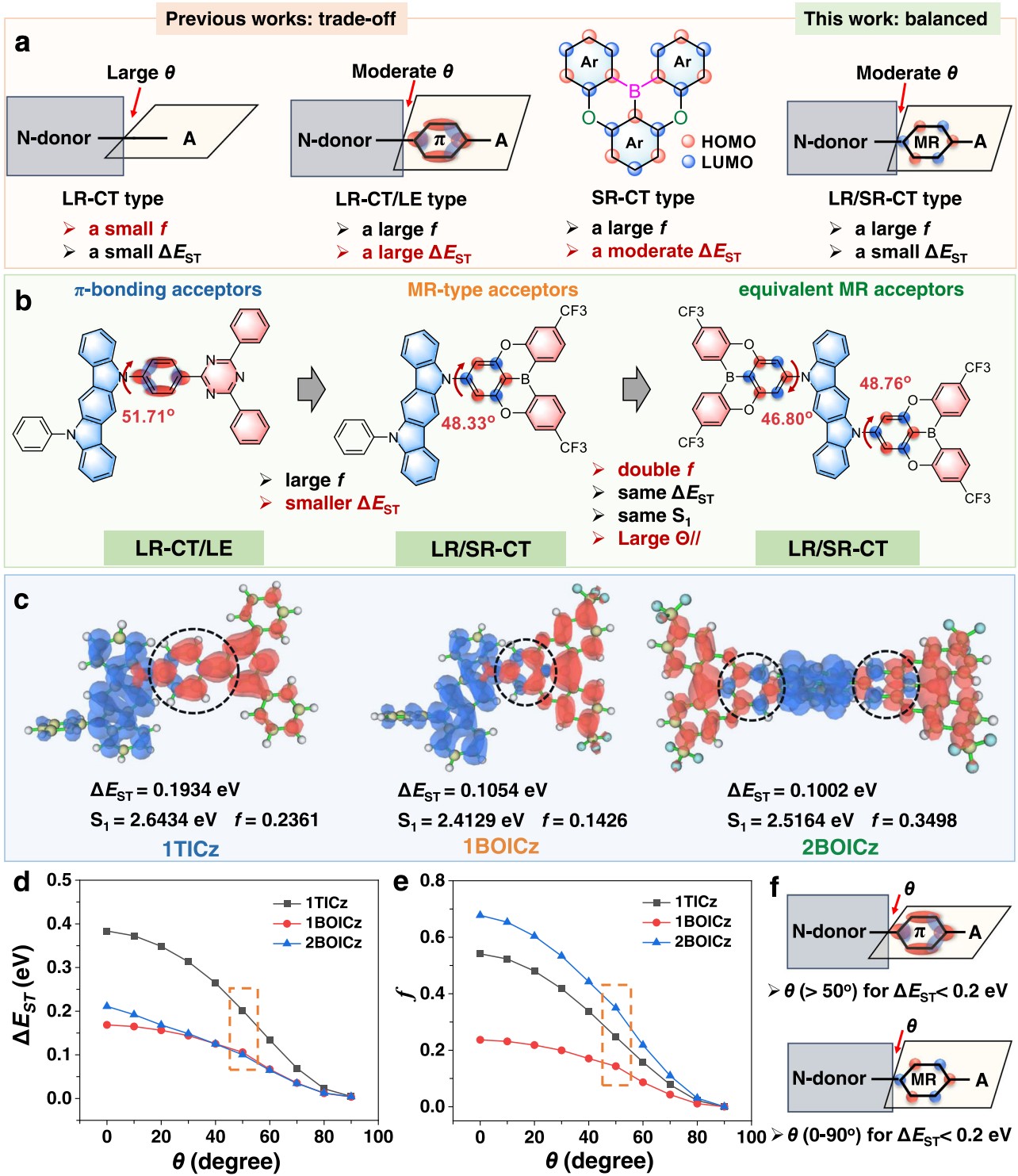

**Fig. 1 | Schematic illustration of TADF molecular design strategy. a** The design principle for TADF emitters with different electronic excitations. **b** The optimized molecular structure of 1TICz (control molecule), 1BOICz and 2BOICz. **c** The distributions of the HOMO (blue color) and LUMO (red color) and the calculated energy levels. **d**, **e** DFT-calculated values of $\Delta E_{ST}$ and $f$ as a function of varied $\theta$ between donor and acceptor at the B3LYP/6-31 G(d, p) level. The dashed box corresponds to $\theta$ of a DFT-optimized structure for three TADF emitters in the ground state. **f** Effect of the varied $\theta$ between donor and acceptor on $\Delta E_{ST}$ for emitters with conventional acceptor and MR acceptor.

the $f$ value but only slightly enlarges the $\Delta E_{ST}$ which remained at a moderate value of ~0.16 eV even at $\theta = 0$. As referred from the orbital distributions, hybrid LR/SR-CT excitations were observed at a small or moderate $\theta$, suggesting that this unique excitation type can well balance a small $\Delta E_{ST}$ and a large $f$ value in a large $\theta$ region. But when a large $\theta$ is adopted, the extension of HOMO from the donor to the phenyl linkage has vanished and only LR-CT excitations can be obtained, thus greatly reducing the $f$ value. Therefore, to guarantee the hybrid short- and long-range CT distribution, a sterically uncrowded donor is crucial. In fact, this explains the reason why no LR/SR-CT excitation was observed in previously reported D-A type TADF emitters with BO-acceptors. Following the conventional wisdom, large steric N-donors

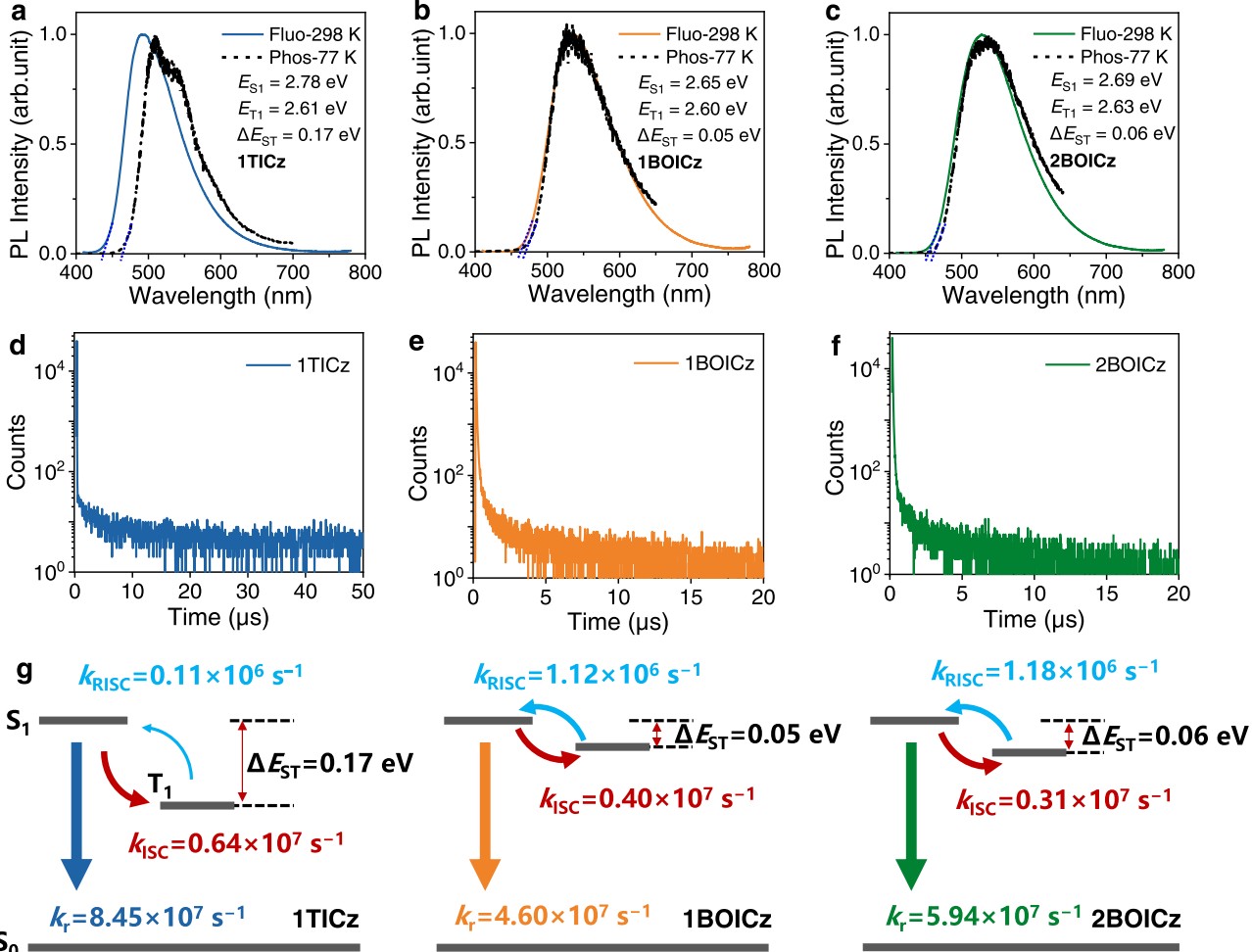

**Fig. 2 | Photophysical properties of 1TICz, 1BOICz, and 2BOICz doped films.**
**a–c** Fluorescence (line, 298 K) and phosphorescence (dash dot, 77 K) spectra of
their doped films (20 wt% doped films). **d–f** Transient PL decay curves of their
doped films (20 wt% doped films). **g** Schematic illustration of decay, up-conversion,
and intersystem crossing processes.

were usually adopted in those emitters and thus the orbital distribu-
tions were totally separated, affording only LR-CT excitations[15,26].
Therefore, only a small $f$ value can be obtained, though the small $\Delta E_{ST}$,
rendering most of those molecules significant efficiency roll-off
though some showed cutting-edge maximum EQEs.

Of particular note, 2BOICz simultaneously exhibited a small $\Delta E_{ST}$
similar to 1BOICz and an even larger $f$ value than 1TICz, successfully
breaking the mutual exclusion between a small $\Delta E_{ST}$ and a large $f$.
Those findings provide a screen that one can construct TADF emitters
with equivalent multiple acceptors (or donors) to generate equivalent
multiple CT channels, which can correspondingly multiply the $f$ value
without influencing the small $\Delta E_{ST}$. To the best of our knowledge,
Adachi et al. unveiled that for TADF emitters with LR-CT/LE excitations,
delocalizing the frontier molecular orbital distributions would benefit
to balance a large $f$ and a small $\Delta E_{ST}$. However, the proof-of-the-
concept emitters they developed only exhibited tens of microsecond-
scale $\tau_D$ and moderate device efficiencies with significant efficiency
roll-off[16,27]. Our molecular design principle proposed here is clearly
distinguished from their strategy and opens a new pathway towards
ideal TADF emitters.

**Photophysical and electronic properties**
The photophysical properties of both target emitters were studied in
toluene with a concentration of $10^{-5}$ M as depicted in Supplementary
Fig. 4 and Supplementary Table 2. Similar ultraviolet-visible (UV)

absorption spectra were observed for both compounds, with strong,
high-energy narrowband absorption peaking at 323 nm, 339 nm,
383 nm, and 403 nm, while wide, relatively weak absorption bands at
425 nm. The former should arise from the intrinsic n-π* or π-π* tran-
sition of acceptor and donor units while the latter can be assigned to
the intramolecular CT transitions. The fluorescent spectra exhibited
wide and structureless green emission with peaks at 527 nm and
518 nm for 1BOICz and 2BOICz, respectively. The onset of UV-
absorption spectra also defined the optical energy gap ($E_g$) of
1BOICz (2.63 eV) and 2BOICz (2.64 eV). Meanwhile, we determined the
HOMO energy levels of both compounds by ultraviolet photoemission
spectroscopy (UPS) in pure neat films as illustrated in Supplementary
Fig. 6. HOMO levels were estimated to be −5.89 eV for 1BOICz and
−6.02 eV for 2BOICz and the LUMO energy levels thereafter can be
deduced from the HOMO and $E_g$ to be −3.26 and −3.38 eV for 1BOICz
and 2BOICz, respectively.

The TADF characters of both emitters were fully characterized by
being dispersed in 9-(3-(9H-carbazol-9-yl)phenyl)−9H−3,9′-bicarba-
zole (mCPBC) matrix with a concentration of 20 wt%[28]. For compar-
ison, the properties of 1TICz in the same conditions were also
evaluated. Figure 2a–c provided the fluorescence and phosphores-
cence spectra of all doped films and wide structureless emissions were
observed, indicating the CT characters of their $S_1$ and $T_1$. Therefore, the
spectra onset defines the energies of $S_1$ and $T_1$, being 2.78 eV and
2.61 eV for 1TICz, 2.65 eV and 2.60 eV for 1BOICz and 2.71 eV and

**Table 1 | Photophysical properties of 1TICz, 1BOICz, and 2BOICz in doped films**

| Compound | $\lambda_{PL}$ (nm)[a] | $E_{S1}$ (eV)[b] | $E_{T1}$ (eV)[b] | $\Delta E_{ST}$ (eV)[b] | $\Phi_P$ (%)[c] | $\Phi_D$ (%)[c] | $\tau_P$ (ns)[d] | $\tau_D$ (ns)[d] | $k_r$ ($10^7$ s$^{-1}$)[e] | $k_{ISC}$ ($10^7$ s$^{-1}$)[e] | $k_{RISC}$ ($10^6$ s$^{-1}$)[e] |
|---|---|---|---|---|---|---|---|---|---|---|---|
| 1TICz | 495 | 2.78 | 2.61 | 0.17 | 0.93 | 0.07 | 11 | 9889 | 8.45 | 0.64 | 0.11 |
| 1BOICz | 534 | 2.65 | 2.60 | 0.05 | 0.92 | 0.08 | 20 | 968 | 4.60 | 0.40 | 1.12 |
| 2BOICz | 528 | 2.69 | 2.63 | 0.06 | 0.95 | 0.05 | 16 | 884 | 5.94 | 0.31 | 1.18 |

[a]20 wt% doped in mCPBC matrix.
[b]Singlet ($E_{S1}$) and triplet ($E_{T1}$) energy levels, $\Delta E_{ST} = E_{S1} - E_{T1}$.
[c]Fractional quantum yields for prompt ($\Phi_P$) and delayed fluorescence ($\Phi_D$).
[d]Lifetime for prompt ($\tau_P$) and delayed ($\tau_D$) fluorescence.
[e]Rate constant of fluorescence radiative decay ($k_r$), intersystem crossing ($k_{ISC}$), and reverse intersystem crossing ($k_{RISC}$), $k_r = \Phi_P/\tau_P$, $k_{ISC} = (1-\Phi_P)/\tau_P$, $k_{RISC} = \Phi_D/(k_{ISC}.\tau_P.\tau_D.\Phi_P)$.

2.65 eV for 2BOICz, respectively. Compared with the much larger $\Delta E_{ST}$ value of 1TICz (0.17 eV), strikingly small values of 0.05 eV for 1BOICz and 0.06 eV for 2BOICz were obtained. Those results are consistent with the theoretical results aforementioned, suggesting that the smaller $\Delta E_{ST}$ values of 1BOICz and 2BOICz arise from their limited HOMO-LUMO overlap by the MR-type orbital distribution on the bridge-phenyl rings.

The PL decay curves and temperature-dependent decay spectra of the three doped films were also measured under an excitation wavelength of ~365 nm as illustrated in Fig. 2d–f and Supplementary Fig. 8, all exhibiting clear TADF behaviors with both prompt and delayed components. Interestingly, unlike most TADF emitters, only rather weak delayed parts were observed and over 90% were from the prompt components. The large ratio of the prompt part evidenced that most excitons are directly radiative decay to the ground states rather than to the triplet states via ISC process. Combining with the prompt PL efficiency ($\Phi_P$) and lifetimes ($\tau_P$) of those three compounds (Table 1), large $k_r$ values of $8.45 \times 10^7$ s$^{-1}$, $4.60 \times 10^7$ s$^{-1}$ and $5.94 \times 10^7$ s$^{-1}$ can be recorded for 1TICz, 1BOICz, and 2BOICz, respectively. The corresponding $k_{ISC}$ values of $0.64 \times 10^7$ s$^{-1}$ for 1TICz, $0.40 \times 10^7$ s$^{-1}$ for 1BOICz and $0.31 \times 10^7$ s$^{-1}$ for 2BOICz were also obtained. For all compounds, the $k_r$ is even tenfold higher than $k_{ISC}$. This situation is rare for most donor-acceptor type TADF emitters, which should arise from their large $f$ values. In terms of the delayed components, different from the microsecond-scale delayed lifetime of 1TICz (9.89 μs), sub-microsecond-scale delayed components were observed for 1BOICz (0.97 μs) and 2BOICz (0.88 μs), respectively. The $k_{RISC}$ values of those three compounds were obtained to be $0.11 \times 10^6$ s$^{-1}$, $1.12 \times 10^6$ s$^{-1}$, and $1.18 \times 10^6$ s$^{-1}$ for 1TICz, 1BOICz, and 2BOICz, respectively. The $k_{RISC}$ of 1TICz agrees with previous reports and the relatively smaller value should be due to its larger $\Delta E_{ST}$. Meanwhile, the emitters with MR acceptors showed nearly five times larger $k_{RISC}$ values, which naturally benefit from their small $\Delta E_{ST}$s.

Interestingly, though their $k_{RISC}$ values are not the cutting-edge ones, $\tau_D$s of 1BOICz and 2BOICz are even shorter than the ones with $k_{RISC} > 10^7$ s$^{-1}$[29,30]. As aforementioned, in addition to $k_{RISC}$, the competition between $k_r$ and $k_{ISC}$ is also the decisive factor that controls the exciton lifetimes. The rate constants of the three compounds are illustrated in Fig. 2g. For the three compounds, the $k_r$ values are over tenfold higher than $k_{ISC}$ values and thus most S$_1$ will directly decay to the ground state, rather than repeating the S$_1 \leftrightarrow$ T$_1$ spin-flip transition cycles. Under this circumstance, the $k_{RISC}$ is the true rate-determining process to control exciton lifetimes and a rate constant in the range of $10^6$ s$^{-1}$ will lead to a sub-microsecond delayed lifetime, as is the case of 1BOICz and 2BOICz. And the even shorter $\tau_D$ of 2BOICz should arise from its larger $k_r$, which further reduces the spin-flip transitions compared with 1BOICz. In comparison, the inefficient $k_{RISC}$ of 1TICz generated a much longer delayed lifetime. Those results validate that $k_r \gg k_{ISC} \sim k_{RISC} > 10^6$ s$^{-1}$ is an effective dynamic model to achieve a sub-microsecond-scale delayed lifetime. Different from previous works that pursue extremely large $k_{RISC}$ values, our work here provides an alternative strategy to shorten the delayed lifetime of TADF emitters.

Notably that similar to 1TICz, previous works have revealed other TADF emitters possessing $k_r$ larger than $k_{ISC}$ when adopting a carbazole similar to sterically uncrowded donors[31]. However, the $k_{RISC}$ values of those emitters are rather slow owing to the large $\Delta E_{ST}$. Taking the most representative DACT-II as an example[9], an extremely large $k_r$ of approaching $10^8$ s$^{-1}$ has been obtained, which is also tenfold higher than $k_{ISC}$, but a much slow $k_{RISC}$ in the order of $<10^5$ s$^{-1}$. Based on the above findings, we envision that simply adopting an MR acceptor for such classic TADF emitters may unlock the full potential of their performances.

It should also be pointed out that the MR acceptor with *para*-positioned CF$_3$ substituents on the B atom was first adopted among BO derivations. To illustrate the role of CF$_3$, we have also constructed a reference compound (2BOICz-tBu) using 2BOICz as the model molecule while a previously reported BO-acceptor with tert-butyl groups on the *meta*-positions of B atom was adopted. The calculated electronic properties, synthesis procedure, and characterizations, and photophysical behaviors of 2BOICz-tBu were provided in the Supplementary Information. As illustrated in Supplementary Fig. 9, a similar hybrid orbital distribution was also observed for 2BOICz-tBu, suggesting that our molecular design is a universal one. But clear differences can also be observed. For 2BOICz, the orbitals on the attached phenyl rings can be well constrained on single atoms and thus a small overlap integral (0.2950) for a small $\Delta E_{ST}$ (0.1002). On the contrary, in terms of 2BOICz-tBu, some LUMOs on the bridge-phenyl rings would extend to the adjacent atoms where HOMOs are located, thus enlarging the orbital overlap (0.3045) for a relatively larger $\Delta E_{ST}$ (0.1270). This discrepancy proved that the existence of CF$_3$ groups could enhance the MR-type LUMO distributions on the attached phenyl rings. Besides, owing to the weak electron-withdrawing ability of the tert-butyl units substituted BO-acceptor, 2BOICz-tBu showed a significantly blue-shifted fluorescence emission with a high S$_1$ energy of 2.90 eV. Moreover, a well-resolved phosphorescence spectrum resembling that of the donor segment was recorded, suggesting its LE character of triplet state with an energy of 2.65 eV. A large $\Delta E_{ST}$ of 0.25 eV was thereafter obtained, affording rather poor TADF properties. Contrarily, adopting the CF$_3$ substituted on acceptors of 2BOICz greatly reduced the energy of its CT singlet and CT triplet states, avoiding the influence of low-lying LE triplet and thus favoring a small $\Delta E_{ST}$ for efficient RISC.

Based on those findings, we found that serval criteria should be satisfied to maximize the performances of TADF emitters with LR/SR-CT excitations. Firstly, a moderate dihedral angle between the donor and MR acceptor segments is necessary to guarantee HOMO extension to the bridge-phenyl rings. Secondly, the LE triplet states of the donor and MR acceptor should be energetically close to or higher than CT states to avoid its influence on $\Delta E_{ST}$s[32]. Thirdly, proper substitutions on MR-acceptors should be considered to enhance the MR characters to guarantee the SR-CT excitations on the bridge-phenyl rings. It should be pointed out that some cutting-edge molecules in literature with mainly LR-CT transitions have also realized a short-delayed lifetime of <1 μs by modulating the energy levels of LE triplet (³LE) and CT singlet (¹CT) excited states[29,33,34]. Our molecular design here is totally different

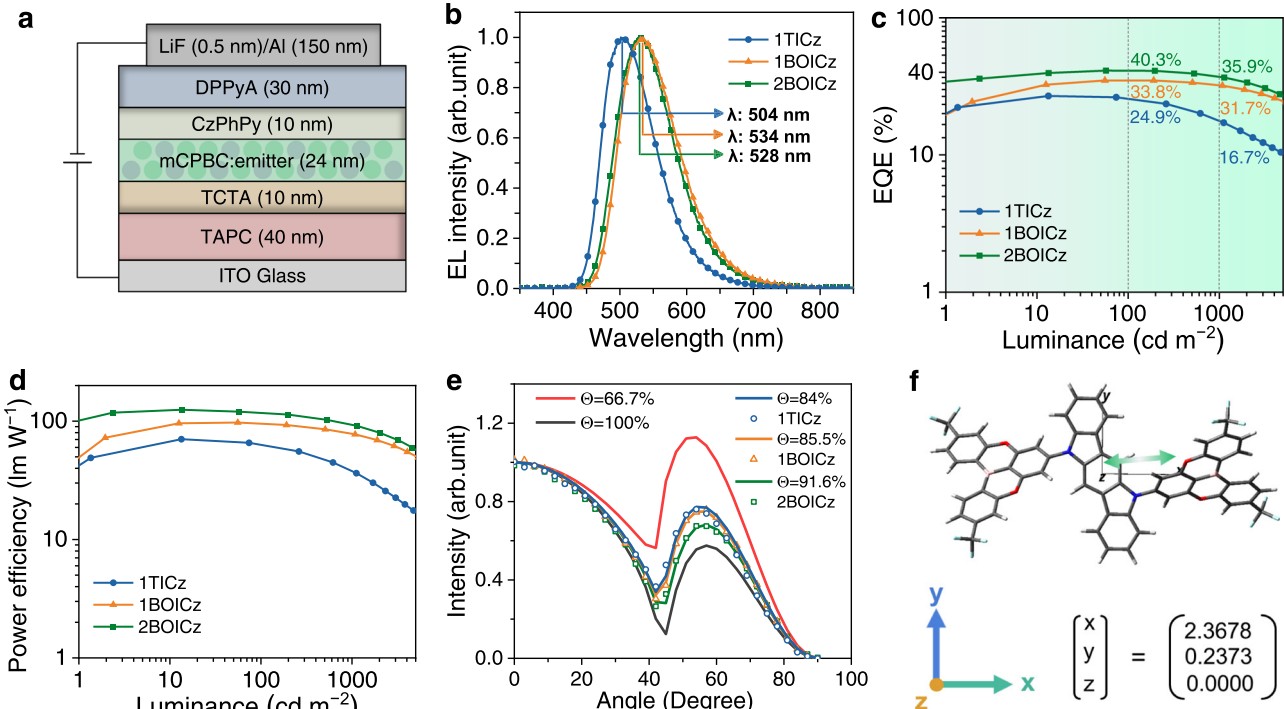

**Fig. 3 | Device structure and performance of OLEDs. a** Architectures of the devices. **b** Normalized EL spectra under 1000 cd m$^{-2}$. **c** EQE versus luminance characteristics. **d** Power efficiency versus luminance characteristics. **e** Angle-dependent PL spectra of 2BOICz, 1BOICz and 1TICz doped in mCPBC host. **f** The direction of the calculated $S_0$-$S_1$ transition dipole moment (as indicated by the arrow) of 2BOICz.

with previous works as what we tried to manipulate is the orbital distribution type. And those previously reported strategies to enhance the RISC process should also work for our emitters, which may further improve the performances of LR/ST-CT emitters.

**Device characterization and performance**

The electroluminescence (EL) performances of both emitters were further evaluated using the following device architecture: Indium tin oxide (ITO)/TAPC (4,4′-cyclohexylidenebis[N,N-bis(4-methylphenyl) benzenamine]) (40 nm)/TCTA (tris(4-(9H-carbazol-9-yl)phenyl)amine) (10 nm)/mCPBC: emitters (24 nm)/CzPhPy (4,6-bis(3-(9H-carbazol-9-yl)phenyl)pyrimidine) (10 nm)/DPPyA (9,10-bis(6-phenylpyridin-3-yl) anthracene) (30 nm)/LiF (lithium fluoride) (0.5 nm)/Al (150 nm). The concentrations of those emitters were optimized to be 20 wt% (Supplementary Table 3–5) and the diagram of the device architectures is illustrated in Fig. 3a. For comparison, the control device with 1TICz was constructed. The EL spectra recorded at 1000 cd m$^{-2}$ were provided in Fig. 3b and showed wide emission with peaks at 534 nm for 1BOICz, 528 nm for 2BOICz, and 504 nm for 1TICz, corresponding to Commission Internationale de l´Eclairage (CIE) coordinates of (0.381, 0.566), (0.383, 0.550) and (0.247, 0.499), respectively, which consisted with the PL spectra of the doped films, indicating the complete energy transfer. Supplementary Figs. 12–17 revealed an interesting phenomenon that both 1BOICz and 2BOICz-based devices exhibited obviously red-shifted emissions with increased dopant concentrations while those one with 1TICz only showed limited red-shift. This can be well explained by the much larger excited-state dipole moments of 1BOICz (1.11 D) and 2BOICz (1.16 D) than that of 1TICz (0.37 D), induced by the strong electron-withdrawing ability of CF$_3$ substituted BO-acceptors.

The EQE versus luminance plots of those devices were illustrated in Fig. 3c, a significantly improved EQE$_{max}$ up to 34.6% was obtained for 1BOICz, much higher than that of 1TICz (26.1%). A greatly alleviated efficiency roll-off was also noted for 1BOICz-based device, with EQE values of 33.8% and 31.7% at high luminance of 100 cd m$^{-2}$ and

1000 cd m$^{-2}$. On the contrary, the EQE of 1TICz-based device sharply decreased to 24.9% and 16.7% at 100 cd m$^{-2}$ and 1000 cd m$^{-2}$. The differences in efficiency roll-off behaviors were believed to be originated from the TADF emissive dynamics of the two emitters under electrical excitation, which will be depicted later by EL decay curves. Notably, 2BOICz-based device showed an exceptionally high EQE$_{max}$ of 40.4%, calibrated using the angle-dependent EL distribution (Supplementary Fig. 18), which was maintained at 40.3% and 35.9% at 100 cd m$^{-2}$ and 1000 cd m$^{-2}$, respectively. A maximum power efficiency (PE$_{max}$) of 122.4 lm W$^{-1}$ was also observed for 2BOICz as illustrated in Fig. 3d, obviously outperforming those of 1BOICz (95.5 lm W$^{-1}$) and 1TICz (71.7 lm W$^{-1}$), respectively. We have also measured the angle-dependent EL intensities of the devices based on 1TICz, 1BOICz, and 2BOICz, and nearly Lambertian profiles pattern with the Lambertian coefficients of 0.97, 0.98 and 0.98 were recorded (Supplementary Fig. 18), respectively. Especially, the calibrated EQE$_{max}$ value of 2BOICz device was still up to 40%, indicating that the high device efficiency was not overestimated. To confirm the reproducibility of the peak EQE$_{max}$ of the device based on 2BOICz, ten groups of devices (40 testing points) were fabricated in parallel. The statistics of those EQE$_{max}$s show that the device performance is in good reproducibility.

To better understand the origin of the impressive efficiencies of those devices, we investigated the angle-dependent $p$-polarization-resolved PL intensity measurement of the corresponding emitting layers. The varied PL intensity measured with different angles was given in Fig. 3e and was analyzed with the classical dipole optical simulations. A high ratio of horizontal emitting dipole orientation ($\Theta_{//}$) of 85.5% and 84.0% were obtained for 1BOICz and 1TICz, which was further increased to 91.6% for 2BOICz. Owing to the moderate dihedral angles between donor and acceptors, those molecules possess large quasi-planar structures, which were further enlarged by the dual equivalent acceptors. The large molecular planarity should render the molecular orientation parallel to the base plane during the evaporation process. Meanwhile, the calculated $S_0$-$S_1$ transition dipole moment as

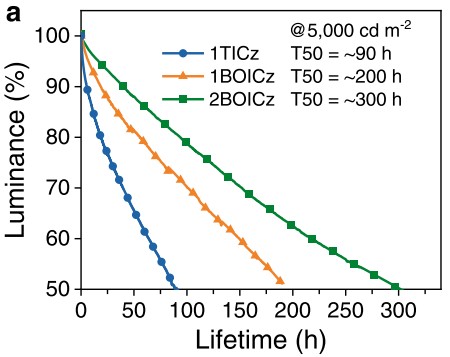
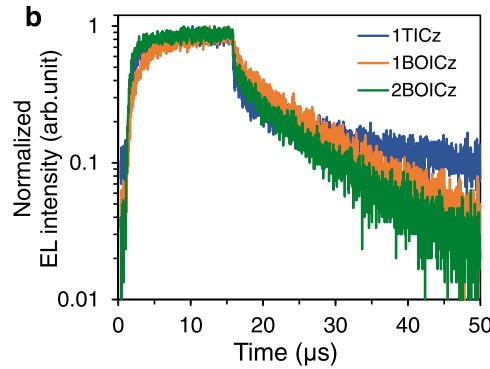

**Fig. 4 | Stability and EL transient characteristics of TADF devices. a** The operational lifetimes of the devices, and **b** EL transient spectra of devices based 1TICz, 1BOICz, and 2BOICz.

illustrated in Fig. 3f and Supplementary Fig. 19 is nearly parallel to the molecular orientation for all three compounds, thus generating those high $\Theta_{//}$s. Previous work has validated the critical role of a high $\Theta_{//}$ in enhancing the light outcoupling efficiency for OLEDs[35–37]. The highest $\Theta_{//}$ of 2BOICz among the three compounds also accounts for its highest efficiency. Besides, the low refractive index of the hole-transporting material, TAPC, is also beneficial to this high EQE. As a comparison, we have also fabricated a device with another most commonly used hole-transporting material, NPB, which possesses a relatively higher refractive index, and a relatively lower $EQE_{max}$ of 38% was realized (Supplementary Fig. 20). Therefore, the excellent TADF properties, a high $\Theta_{//}$ and an optimized device structure should concurrently account for the cutting-edge $EQE_{max}$ of 2BOICz. This is in accordance with the theoretically predicted EQE efficiency performed by commercial software OFSS 1.0[38,39] (Wuhan Yuwei Optical Software Co., Ltd.) (Supplementary Fig. 21). In fact, previous works have also demonstrated $EQE_{max}$s of over 40% using TADF emitters with similar $\Phi$ and $\Theta_{//}$ values as illustrated in Supplementary Table 7. It is interesting to note that most of those previously reported emitters showed relatively larger efficiency roll-off compared with 2BOICz, though their similar $EQE_{max}$s. This further evidenced the advantages of our emitters with hybrid LR/SR-CT excitations to achieve a short-delayed component for an alleviated efficiency roll-off.

We further evaluated the operational stabilities of those devices at a constant current with an initial luminance of 5000 cd m$^{-2}$ as shown in Fig. 4a. Decent half-lifetimes (LT50s) of 89 h, 196 h and 302 h were obtained for 1TICz, 1BOICz, and 2BOICz-based devices, respectively. Using a commonly adopted degradation acceleration factor (n) of 1.75, a LT50 at an initial luminance of 1000 cd m$^{-2}$ can be extrapolated with the equation of LT50 (1000 cd m$^{-2}$) = LT50 (5000 cd m$^{-2}$) × (5000 cd m$^{-2}$/1000 cd m$^{-2}$)$^n$, being 1486 h for 1TICz, 3272 h for 1BOICz and 5043 h for 2BOICz, respectively. Notably that the longest lifetime was observed for 2BOICz, over three times longer than that of 1TICz, evidencing that the molecular design strategy possesses the potential to improve device stability compared with the conventional ones[40].

To understand the underlying physics of the alleviated efficiency roll-off and extended operational stability of the two novel emitters under electrical excitation, the EL decay curves of those devices under an operation voltage of 6 V were recorded. It should be further noted that different to the situation in PL excitations, 75% of the generated excitons under EL excitations would be triplet ones. Therefore, the ratio of the delayed component will greatly be enlarged and a balance in $k_r$, $k_{ISC}$, and $k_{RISC}$ to short $\tau_D$ would matter more for device performances. As provided in Fig. 4b, the $\tau_D$s of all three compounds showed a trend of 1TICz > 1BOICz > 2BOICz, which is exactly inverse to their LT50s. As aforementioned, various bimolecular annihilation processes have been acknowledged as the dominant reasons for not only the efficiency roll-off under high luminance but also the operation

stability[11,41]. Briefly, those bimolecular annihilation processes strongly depend on both concentrations and residence time of excitons. The small efficiency roll-off and the longer LT50s of both 1BOICz and 2BOICz should arise from their shorter delayed lifetimes than 1TICz, which can greatly suppress exciton annihilations under EL excitation.

Besides being directly adopted as emitters, TADF molecules have also been widely used as sensitizers for narrowband MR emitters to assist the recycling of triplet states, known as hyperfluorescence or TADF-sensitized fluorescence (TSF). As an ideal sensitizer, a fast RISC process is required to enhance exciton utilization and a quick radiative decay is also necessary to guarantee the desired Förster energy transfer (FET). The well-balanced $k_r$ and $k_{RISC}$ of the LR/SR-CT type TADF molecules, therefore, render them good sensitizers for MR emitters. To validate this inspiration, the performances of 1BOICz and 2BOICz as sensitizers were evaluated using DBN-ICz as the MR emitter[42]. For comparison, 1TICz was also adopted as a sensitizer. The detailed photophysical properties and energy transfer processes were given in Supplementary Fig. 23. All three sensitizers provided efficient spectral overlapping between their emission spectra with the absorption spectrum of DBN-ICz emitter, affording large FET radius ($R_0$) of 3.4, 3.3, and 3.3 nm for 1TICz, 1BOICz, and 2BOICz-based systems, respectively, implying the efficient energy transfer in these sensitizing systems. TSF-OLEDs were fabricated with mCPBC: 20 wt% sensitizers: 1 wt% DBN-ICz as EMLs while all the other functional layers were similar to the TADF devices. Figure 5a provides the EL spectra of those three devices, all showing narrowband emissions with a peak at 551 nm, a small FWHM of 25 nm, and CIE coordinates of (0.37, 0.61). Clear differences were observed from their EQE-luminance characteristics as displayed in Fig. 5b. The 1BOICz and 2BOICz-based TSF devices achieved high $EQE_{max}$ values of 36.6% and 37.6%, respectively, with greatly reduced efficiency roll-off (Table 2). On the contrary, 1TICz-based TSF device displayed not only a relatively lower $EQE_{max}$ (32.2%) but also a larger efficiency roll-off. The EL decay curves as illustrated in Supplementary Fig. 24 and depicted relatively shorter lifetimes for devices based on 1BOICz and 2BOICz, which should benefit to suppress exciton annihilations under high luminance and thus lower efficiency roll-off than the device using 1TICz. The quick exciton consumption of 1BOICz and 2BOICz-based devices should also be assigned to the more balanced $k_{RISC}$ and $k_r$ from the LR/SR-CT orbital distributions.

## Discussion

In summary, a novel LR/SR-CT type TADF molecule design principle was proposed, breaking the trade-off between a small $\Delta E_{ST}$ and a large $f$ towards the goal of a short $\tau_D$. By attaching equivalent MR acceptors onto a sterically uncrowded nitrogen-donor, this architecture not only creates a hybrid orbital distribution involving a hybrid donor-to-acceptor LR-CT and bridge-phenyl SR-CT characteristics to balance a

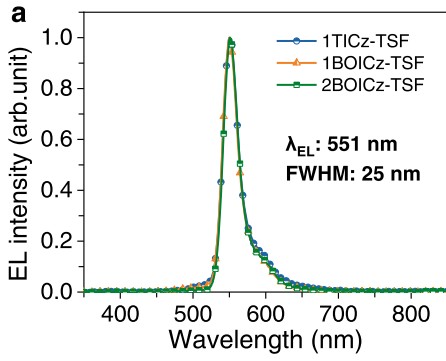

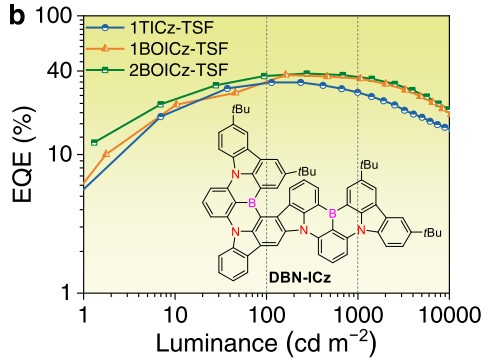

**Fig. 5 | TSF-OLED device performances. a** Normalized EL spectra under 1000 cd m⁻². **b** EQE versus luminance characteristics, and the narrowband yellow MR emitter DBN-ICz.

**Table 2 | Summary of the EL performance of TADF-OLEDs and TSF-OLEDs devices**

| Device | Compound | $\lambda_{EL}$[a] [nm] | FWHM [nm] | $V_{on}$[b] [V] | EQE$_{max/100/1000}$[c] [%] | PE$_{max/100/1000}$[d] [lm W⁻¹] | CIE (x,y)[e] |
|---|---|---|---|---|---|---|---|
| | 1TICz | 504 | 100 | 3.1 | 26.1/24.9/16.7 | 71.7/61.6/37.1 | (0.247, 0.499) |
| TADF[f] | 1BOICz | 534 | 101 | 3.1 | 34.6/33.8/31.7 | 95.5/92.1/76.0 | (0.381, 0.566) |
| | 2BOICz | 528 | 91 | 3.0 | 40.4/40.3/35.9 | 122.4/114.4/89.6 | (0.383, 0.550) |
| | 1TICz | 551 | 25 | 3.0 | 32.2/32.1/26.3 | 88.0/87.4/60.9 | (0.371, 0.611) |
| TSF[f] | 1BOICz | 551 | 25 | 3.0 | 36.6/32.9/34.4 | 99.3/94.5/87.5 | (0.372, 0.612) |
| | 2BOICz | 551 | 25 | 3.0 | 37.6/36.2/35.4 | 102.3/101.2/89.4 | (0.372, 0.613) |

[a]EL peak wavelength.
[b]Turn-on voltage (V$_{on}$).
[c]Maximum external quantum efficiency (EQE), value at 100 and 1000 cd cm⁻².
[d]Maximum power efficiency (PE), value at 100 and 1000 cd cm⁻².
[e]CIE coordinates at 1000 cd cm⁻².
[f]Both PE and EQE values of TADF and TSF devices are calibrated by Lambertian correction.

large $f$ and a small $\Delta E_{ST}$, but also allows equivalent radiative channels to further enlarge the $f$ without affecting the $\Delta E_{ST}$. The proof-of-the-concept emitter, 2BOICz, displayed well-balanced rate constants to establish an exciton dynamic model of $k_r \gg k_{ISC} \sim k_{RISC} \gg 10^6$ s⁻¹ for an extraordinarily fast emission lifetime of 0.88 µs. The quasi-planar structure of this emitter also favors essentially high $\Theta_{//}$ for a large outcoupling efficiency and the corresponding device exhibited a remarkably high EQE$_{max}$ of 40.4%, which remained at 35.9% at 1000 cd m⁻² with an extended LT50 of over 5000 h simultaneously. The balanced fast RISC and radiative decay also render this compound an ideal sensitizer for MR emitter and a high EQE$_{max}$ of 37.6% with alleviated efficiency roll-off was realized with an extremely small FWHM of 25 nm. It is also envisioned that within the prerequisite of maintaining a larger $k_r$ than $k_{ISC}$, the $k_{RISC}$ may be further improved by enlarging the spin-orbital coupling (SOC) by introducing heavy atoms and modulating the energy levels of the localized and CT states. A potential molecular design is to adopt sulfur-bridged boron (BS) MR acceptors. Considering that previous D-A type TADF emitters always bear LR-CT or LR-CT/LE type orbital distributions, we believe that the LR/SR-CT type proposed here should represent a concept advancement towards a new paradigm of TADF emitters with short-delayed components.

## Methods

The synthetic procedures of both boron-based compounds were illustrated in supporting information. The intermediate BO-Br was prepared using the previously reported cyclization reactions in the presence of n-butyllithium (n-BuLi) and boron tribromide (BBr₃). Afterward, the Buchwald-Hartwig amination was conducted to synthesize the final compounds with high yields of over 70%. The final products were further purified using temperature-gradient vacuum sublimation, of which the chemical structures were then fully characterized using ¹H nuclear magnetic resonance (NMR) spectroscopy, mass spectrometry (MS), and elemental analysis. Detailed synthetic procedures and characterization data are provided in the supplementary information.

## Data availability

The data supporting the findings of this study are available within the paper and the Supplementary Information. Source data are provided with this paper.

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

## Acknowledgements

This work was supported by the National Key Basic Research and Development Program of China (Grant no. 2020YFA0715000 and 2017YFA0204501), the National Science Fund of China (Grant No. 22135004, 52222308 and 61890942), the Guangdong Major Project of Basic and Applied Basic Research (Grant no. 2019B030302009), and the Guangdong Basic and Applied Basic Research Foundation (2021B1515120041). We appreciate Professor Honggang Gu at the School of Mechanical Science and Engineering, Linya Chen at School of Optical and Electronic Information, Huazhong University of Science and Technology for support in optical simulations of OLED devices.

## Author contributions

L.D. and D.Z. conceived and supervised the project. D.Z. and L.D. designed the experiments. G.M. and J.Z. synthesized and character-ized the TADF emitters. G.M. performed the theoretical calculations, photophysical characterization, OLED fabrication, and measurement. T.F. and X.Z. helped with the theoretical calculation. X.W. and H.D.

participated in the discussion of the photophysical mechanism. Y.Z. and Q.W. provided suggestions on device fabrication and experiments. D.M. and D.Y. helped in measuring the molecular dipolar orientation. D.Z. and L.D. analyzed the results and wrote the manuscript.

## Competing interests

The authors declare no competing interests.
