## [Peer Review File · Nature Communications]

High-efficiency and stable short-delayed fluorescence emitters with hybrid long- and short-range charge-transfer excitationsREVIEWER COMMENTS

Reviewer #1 (Remarks to the Author):

In this manuscript, authors designed novel D-MR-A molecules to achieve small ΔE_{ST} and large f values. The quasi-planar structure of 2BOICz also favors essentially high Θ for a large outcoupling efficiency and the corresponding device exhibited a record-high EQE_{max} of 41.2% with alleviated efficiency roll-off and extended lifetime. This study shows the novel molecular design and outstanding device efficiency among TADF OLEDs. However, the supportive proof or reason for such high EQE is unclear and should be provided further. I have the following points which need attention before a decision on acceptance can be made.

1. In molecular design, why CF₃ groups are selected and introduced to the para position toward the boron atom?
2. Why use B3LYP/6-31G* to calculate the frontier orbital energy levels and ΔE_{ST} of these molecules? To the best of my knowledge, the values from B3LYP are not accurate when dealing with donor-acceptor systems.
3. In the Supporting Information, the authors mentioned that "The structures were optimized using DFT/single-crystal (S0 state) or TD-DFT (S1 and T1 state) methods with a 6-31G(d,p) basis set" Why the basis is different from the main text? And where are the single-crystal structures for each compound?
4. Page 3, line 85 & Caption of Fig. 1. What are the weights of the HOMO and LUMO contributions for these compounds? I strongly recommend authors to provide the natural transition orbital (NTO) analysis of the S1 and T1 states for these molecules.
5. Page 3, Line 133. The authors mentioned that "The high f of 2BOICz, reminiscent of locally excited (LE) states". What is the "reminiscent of locally excited (LE) states"? The meaning of the phrase is not clear.
6. Caption of Figure S5. "The HOMO levels were determined to be -5.60 eV, -5.98 eV, and -6.02 eV for 1TICz, 1BOICz, and 2BOICz." Please provide the method for obtaining HOMO levels via the ultraviolet photoelectron spectra (UPS). Why 1TICz shows shallower HOMO and LUMO compared to the other two molecules?
7. The authors mentioned that the D-MR-A type TADF molecules 1BOICz and 2BOICz achieved smaller ΔE_{ST} s and larger f values than D- π -A type TADF molecule 1TICz. However, Table S2 shows the PLQYs of 1TICz, 1BOICz, and 2BOICz (90%, 90%, and 89%, respectively). They are very close and do not correlate to the calculated f values. What's the sample condition used for the PLQY measurement? Please provide more discussion.
8. The EQEs of the reported devices exceeded 40% when using 2BOICz. How many devices were fabricated and how was the standard deviation of EQE? Authors should carefully check the reproducibility.

9. Moreover, the ratio of horizontal emitting dipole orientation ($\Theta//$) of 2BOICz was 88% and its PLQY was only 89%. I am not sure the current data can support such a high EQE (41%) via the 2BOICz-based device. Please provide the reason and simulation to support it.

10. Despite the relatively large ΔE_{ST} (0.17 eV) of 1TICz, the delayed lifetime value is only 3222 ns. Do Tn states of 1TICz participate in the RISC process? In addition, the phosphorescence pattern of 1TICz looks dissimilar (Fig 2a) among the three compounds, how to make sure whether it originates from the CT character?

11. In Table S3 to Table S5, as the concentration of 2BOICz-based devices increase from 20 wt% to 40 wt%, the EQEmax slightly decrease from 41.2% to 38.9%. However, why the EL peak wavelengths of 2BOICz are sensitive to the concentrations? It shows a serious red-shifted peak and reveals a high degree of aggregation that may occur when increasing the doping concentration. Please explain.

Reviewer #2 (Remarks to the Author):

The paper deals with new TADF emitters using multiresonance bridge. The new emitters showed a large decay rate over an order of magnitude higher than ISC rate with short-delayed lifetime. However, the B-O accept was already many reported with high efficient TADF materials. Authors should be compared with IBOICz and 2BOICz without electron acceptor CF₃ groups at least in the calculation. The other groups already used {A with MR-D concept}. Therefore, the author should be explained the author's [A-MR-D concept] is different with the already reported [A with MR-D concept]. Although the large decay rate with short-delayed life time, FWHM of IBOICz and 2BOICz showed still broad without MR effect. I did not find the scientific improvement at these points. I do not recommend to publish in the Nature Communication.

Ref) Ahn, D.H. et al. Nat. Photonics 13, 540-546 (2019)

Reviewer #3 (Remarks to the Author):

The authors have claimed that donor-multiresonance bridge-acceptor typed emitters can achieve a small singlet-triplet energy gap and a large oscillator strength. They demonstrated that the D-MR-A emitter showed high efficiency with low roll-off due to a fast RISC rate of over 10^6 /s. However, this reviewer is not positive about supporting the publication of this manuscript in Nature Commun due to the following concerns.

The substantial merit of the MR core in the TADF emitters is a narrow emission spectrum giving a high color purity. Indeed, the typical DABNA-based MR emitter has a very low RISC rate of below 10^4 /s. Therefore, reducing the singlet-triplet energy gap and enhancing the oscillator strength are key issues in

improving the optoelectrical characteristics of OLEDs. In this concern, this reviewer could not find merit in the D-MR-A emitter claimed in this manuscript.

Regarding the fast RICS and low-efficiency roll-off, the D-A or D-A-D typed TADF emitters have already been introduced. The TADF-OLEDs with emitters showed highly efficient performance of over 38% (EQE) with extremely low-efficiency roll-off, a high horizontal dipole ratio of over 90%, and a fast RISC of over 10^6 /s. However, the D-A or D-A-D typed TADF emitters use CT characteristics, which induce broadened EL emission similar to the D-MR-A emitter, resulting in low color purity.

This reviewer agreed that the optoelectrical performance of OLEDs is impressive, however, could not find a novelty in the D-MR-A emitter compared to the D-A-typed TADF emitters in the previous literature. My concern is the D-MR-A emitter seems not to demonstrate the MR characteristics when connected to D and A molecules.

Response to reviewer's comments

Dear reviewers,

We thank all the reviewers for the helpful comments and suggestions, which have helped us to greatly improve our manuscript. We carefully considered all reviewers' comments and did our best to address these concerns. In the text below we provided a point-by-point response to all comments made by the three reviewers. The changes have been marked in the revised Manuscript and Supplementary Information.

REVIEWER COMMENTS

Reviewer #1 (Remarks to the Author):

In this manuscript, authors designed novel D-MR-A molecules to achieve small ΔE_{ST} and large f values. The quasi-planar structure of 2BOICz also favors essentially high Φ_{out} for a large outcoupling efficiency and the corresponding device exhibited a record-high EQEmax of 41.2% with alleviated efficiency roll-off and extended lifetime. This study shows the novel molecular design and outstanding device efficiency among TADF OLEDs. However, the supportive proof or reason for such high EQE is unclear and should be provided further. I have the following points which need attention before a decision on acceptance can be made.

1. In molecular design, why CF_3 groups are selected and introduced to the para position toward the boron atom?

Response: Thank you for your valuable comments.

The adoption of CF_3 groups was aimed at, on one hand, enhancing the electron-withdrawing ability of the BO-acceptors to avoid the influence of the low-lying localized excited (LE) triplet states of the donor; on the other hand, enhancing the MR distributions on the attached phenyl rings for a more balanced f and ΔE_{ST} . It can be clarified as follows.

Firstly, the adoption of CF_3 groups can enhance the electron-withdrawing ability of the BO acceptor. In our work, 5,11-dihydroindolo[3,2-b]carbazole (32bICz) group was adopted as the donor, which possessed a LE triplet state of only 2.65 eV. Therefore, to avoid the influence of LE states on ΔE_{ST} , a strong electron-withdrawing unit should be adopted. We calculated the LUMO levels of the CF_3 -BO unit in our manuscript and other several BO derivatives in previous reports. As illustrated in the figure below, CF_3 -BO showed the deepest LUMO energy, suggesting its strongest electron-withdrawing ability. Using strong acceptors, the energy of the charge transfer (CT) states can be greatly stabilized, facilitating to avoid the influence of the low-lying LE states.

Supplementary Fig. 8 (a, b) The chemical structures of 2BOICz and 2BOICz-tBu, and the calculated values of the dihedral angle (θ), HOMO-LUMO distributions, singlet energy level (S_1) and oscillator strength (f). Furthermore, the overlap integral of the orbitals at S_1 state is calculated by Natural Transition Orbital (NTO) analysis. (c) UV-vis absorption (left axis) in toluene (10^{-5} M), normalized PL (298 K) and phosphorescence (77 K) spectra (right axis) of 2BOICz-tBu in toluene. (d, e) Fluorescence (298 K or 77 K) and phosphorescence (77 K) spectra of doped films (20 wt% doped films).

Based on those findings, we found that several criteria should be satisfied to maximize the performances of TADF emitters with LR/SR-CT excitations. Firstly, a small to moderate dihedral angle between donor and MR-acceptor segments is necessary to guarantee HOMO extension to the bridge phenyl rings. Secondly, the LE triplet states of donor and MR-acceptor should be energetically close to or higher than CT states to avoid its influence on ΔE_{ST} s. Thirdly, proper substitutions on MR-acceptors should be considered to enhance the MR characters to guarantee the SR-CT excitations on the bridge phenyl rings.

In the Supporting Information of the revised manuscript, the detailed study on the properties of 2BOICz-tBu has been provided as a contrast. The synthetic procedure and related characterizations were presented in the Experimental section and Supplementary Figure section. The UV-vis absorption and emission spectra in toluene (10^{-5} M) were shown in **Supplementary Fig. 9a**, the relatively strong absorption strength was observed in the long wavelength, accompanied with a bright deep-blue emission ($\lambda_{PL}=441$ nm, $\Phi_{PL}=85\%$), this result matches well with the calculation that showed a high f value. The solvatochromic effect of 2BOICz-tBu in various solvents further confirmed the strong CT feature in its S_1 state (**Supplementary Fig. 9b**). Furthermore, the 20 wt% 2BOICz-tBu doped in PPF film displayed blue emission with a peak at 466 nm, and an excellent PLQY of 88%. Especially, 2BOICz-tBu revealed a higher ΔE_{ST} of 0.25 eV due to its lower triplet energy compared with that of 2BOICz (ΔE_{ST} of 0.06 eV). Due to the relatively large ΔE_{ST} , 2BOICz-tBu exhibited a negligible TADF behavior in the doped film, only showed a prompt decay lifetime in the

order of nanoseconds (**Supplementary Fig. 9c**).

To evaluate the EL performance of 2BOICz-tBu, we fabricated the OLED devices whose structure is displayed in **Supplementary Fig. 20a**. PPF was used as the host and hole-blocking layer due to its high T_1 (3.1 eV) energy level and shallow HOMO level. As shown in **Supplementary Fig. 20b**, the 20 wt%-40 wt% doped devices exhibited blue EL emission with peaks at ~465 nm and FWHMs of ~56 nm. The current density-voltage-luminance (J-V-L), and EQE, PE, CE versus luminance curves are plotted in **Supplementary Fig. 20c-20f**, and the EL performance parameters were summarized in **Supplementary Table 6**. The devices showed turn-on voltages in the range of 2.9-3.3 V and obtained the highest EQE_{max} value of 17.7%. The EL decay curves of those devices showed long-lasting delayed components, suggesting the TADF characteristic of this emitter though being weak, which accounts for this moderate EQE_{max}. Considering the relatively large ΔE_{ST} , the RISC process of 2BOICz-tBu should arise from the participation of the high-lying triplet excited state (T_n) (DFT calculation section). A high $\Theta_{//}$ factor of 2BOICz-tBu doped PPF film was also obtained to be 89.7%, beneficial to enhance the outcoupling efficiency for an improved EQE. However, it should be pointed out that, 2BOICz-tBu based device showed not only a much lower EQE_{max} but also significant efficiency roll-off compared with the device using 2BOICz. The reason should be attributed to its poor RISC process from its large ΔE_{ST} . All those results provide insight that the CF_3 units on the *para*-position of the BO motif play a critical role in the modulation of excited energies and optoelectronic properties of the investigated compounds in this work.

¹H NMR spectrum of 2BOICz-tBu in CD₂Cl₂.

Mass spectrometry of 2BOICz-tBu.

Supplementary Fig. 7 (a) The ultraviolet photoelectron spectra of the vacuum-deposited thin films for 2BOICz-tBu. The HOMO levels were determined to be -5.50 eV. The band energy is 2.98 eV, the LUMO energy level can be estimated to be -2.52 eV. (b) Decomposition temperature (T_d) with 5% weight loss for 2BOICz-tBu.

Supplementary Fig. 9 (a) UV-vis absorption (left axis) in toluene (10^{-5} M), and normalized PL spectra (right axis) of 2BOICz-tBu in toluene. (b) The solvent-dependent PL spectra. (c) Transient PL decay curve of 20 wt% 2BOICz-tBu doped film. (d) The spin-orbit coupling (SOC) matrix elements between S_1 and T_n states of 2BOICz-tBu were conducted at the B3LYP/6-31G(d,p) level.

Supplementary Fig. 20 (a) Device architecture for 2BOICz-tBu based TADF-OLEDs. (b) EL spectra of the devices under 1000 cd m⁻². (c) Luminance-voltage-current density characteristics. (d) External quantum efficiency (EQE), (e) current efficiency (CE) and (f) power efficiency (PE) versus luminance. (g) EL transient spectra for the 20 wt%, 30 wt% and

40 wt% 2BOICz-tBu doped devices. (h) Angle-dependent PL spectra of 20 wt% 2BOICz-tBu in PPF host.

Supplementary Table 6 Summary of the EL performance of 2BOICz-tBu based devices.

Doping concentration [nm]	$\lambda_{\text{EL}}^{\text{a}}$ [nm]	V_{on}^{b} [V]	$L_{\text{max}}^{\text{c}}$ [cd m^{-2}]	$\text{CE}_{\text{max}/100}^{\text{d}}$ [cd A^{-1}]	$\text{PE}_{\text{max}/100}^{\text{e}}$ [lm W^{-1}]	$\text{EQE}_{\text{max}/100}^{\text{f}}$ [%]	CIE (x,y) ^g
20 wt%	467	3.3	3987	27.5/15.6	27.0/10.2	17.7/10.1	(0.157, 0.201)
30 wt%	468	3.0	5200	21.0/15.3	22.0/10.0	15.1/9.8	(0.155, 0.214)
40 wt%	469	2.9	6589	18.8/15.9	18.5/10.8	12.5/9.8	(0.156, 0.212)

^a EL peak wavelength; ^b Turn-on voltage (V_{on}); ^c Maximum luminescence (L); ^d Maximum current efficiency (CE), value at 100 cd cm^{-2} ; ^e Maximum power efficiency (PE), value at 100 cd cm^{-2} ; ^f Maximum external quantum efficiency (EQE), value at 100 cd cm^{-2} ; ^g CIE coordinates at 1000 cd cm^{-2} .

2. Why use B3LYP/6-31G* to calculate the frontier orbital energy levels and ΔE_{ST} of these molecules? To the best of my knowledge, the values from B3LYP are not accurate when dealing with donor-acceptor systems.

Response: Thank you for your valuable comments.

After comparing with other functional by calculation, we found that the B3LYP functional was a more appropriate basis in our manuscript. Our work here tried to provide an advanced molecule design that breaks the trade-off between a small ΔE_{ST} and a large f value. Therefore, the precise prediction of a small ΔE_{ST} is crucial. Using B3LYP functional, small ΔE_{ST} values of 0.19 eV, 0.11 eV, and 0.10 eV are predicted for 1TICz, 1BOICz and 2BOICz, respectively, which are in fairly good agreement with the experimentally determined ΔE_{ST} values (0.17 eV for 1TICz, 0.05 eV for 1BOICz, and 0.06 eV for 2BOICz). For comparison, we have also calculated the ΔE_{ST} values at the levels of CAM-B3LYP, B97XD, PBE0, and M062X, respectively, and the experimental ΔE_{ST} s are provided in **Supplementary Table**. Clearly, in comparison with the deviation values, the B3LYP functional does a better job than all the other basis. Meanwhile, we noticed that most previously reported donor-acceptor (D-A) TADF emitters based on oxygen-bridged boron acceptors also calculated the energy gaps at the B3LYP/6-31G(d,p) level (*Nat. Photonics* **2019**, *13*, 540). It is also deserved to mention that, on the same basis, both 1BOICz and 2BOICz showed relatively smaller ΔE_{ST} s than 1TICz, irrespective of the basis adopted. Those results further validated our idea.

Supplementary Table The S_1 and T_1 energies as well as ΔE_{ST} computed at the TD-DFT/6-31g(d,p) levels of theory for **1TICz**, **1BOICz** and **2BOICz**. A comparison between computed and experimental ΔE_{ST} is provided.

Compounds	Method	S_1/eV	T_1/eV	$\Delta E_{ST}/eV$	ΔE_{ST}^a
1TICz	TD-CAM-B3LYP	3.76	2.73	1.03	0.17
	TD- B97XD	3.56	2.86	0.70	
	TD-B3LYP	2.64	2.45	0.19	
	TD-PBE0	2.85	2.56	0.29	
	TD-M062X	3.64	3.10	0.54	
1BOICz	TD-CAM-B3LYP	3.61	2.74	0.87	0.05
	TD- B97XD	3.38	2.86	0.52	
	TD-B3LYP	2.42	2.31	0.11	
	TD-PBE0	2.64	2.50	0.14	
	TD-M062X	3.47	3.13	0.34	
2BOICz	TD-CAM-B3LYP	3.65	2.74	0.91	0.06
	TD- B97XD	3.34	2.87	0.47	
	TD-B3LYP	2.52	2.42	0.10	
	TD-PBE0	2.74	2.60	0.14	
	TD-M062X	3.53	3.19	0.34	

^aExperimental value.

3. In the Supporting Information, the authors mentioned that “The structures were optimized using DFT/single-crystal (S_0 state) or TD-DFT (S_1 and T_1 state) methods with a 6-31G(d,p) basis set” Why the basis is different from the main text? And where are the single-crystal structures for each compound?

Response: We are truly sorry for this mistake and the errors have been fixed in the revised Supplementary Information.

We now changed the sentence to “The ground-state structures were fully optimized by the B3LYP functional with the 6 31G(d,p) basis set. The S_n and T_n energies and spin-orbit couplings (SOC S_n - T_n) were calculated using a TD-DFT approach at the B3LYP/6-31G(d,p) level. The natural transition orbital (NTO) analysis of S_1 and T_1 states was carried out using the Multiwfn software.^[5] The SOC matrix elements are computed using the ORCA package^[6].”

Reference:

[5] T. Lu and F. Chen, J. Comput. Chem., 2012, 33, 580.

[6] F. Neese, Wiley Interdiscip. Rev.: Comput. Mol. Sci. 2012, 2, 73-78.

4. Page 3, line 85 & Caption of Fig. 1. What are the weights of the HOMO and LUMO contributions for these compounds? I strongly recommend authors to provide the natural transition orbital (NTO) analysis of the S_1 and T_1 states for these molecules.

Response: Thank you for your valuable comments.

The weights of the HOMO and LUMO contributions for these compounds have been provided in **Supplementary Fig. 1** and ratios of 70.2%, 70.4% and 70.3% have been calculated for 1TICz, 1BOICz and 2BOICz, respectively.

Supplementary Fig. 1 Distributions of HOMO and LUMO of 1TICz, 1BOICz and 2BOICz.

Besides, according to your suggestions, we have provided the natural transition orbitals (NTOs) analysis of the S_1 and T_1 states for these molecules as shown in **Supplementary Fig.2**. It was clear to note that both S_1 and T_1 states of all three emitters were dominated by hole \rightarrow electron transitions with main CT characters. And in agreement with its SR-CT character on the bridge-phenyl ring, 1BOICz possessed a relatively larger ratio of CT transition than 1TICz. Owing to the dual acceptors, 2BOICz showed a relatively lower ratio of CT transitions than 1BOICz, but still higher than 1TICz. Those results further validated our inspirations that the hybrid LR/SR-CT excitation possessed obvious advantages in separating orbital distribution than the hybrid LR-CT/LE excitation, that is a smaller ΔE_{ST} . Further taking its decent f into consideration, the hybrid LR/SR-CT excitation could break the trade-off between a small ΔE_{ST} and a large f .

Supplementary Fig. 2 Natural transition orbitals (NTOs) describing the excitation characters of the S_1 , T_1 and T_2 state in 1TICz, 1BOICz and 2BOICz; the weights of the hole-electron contributions to the excitations are included. For these three compounds, the transition of both S_1 and T_1 states is dominated by hole \rightarrow electron ($> 92\%$) with CT characters, and along with a partially intramolecular short-range CT excitation character on the phenylene.

5. Page 3, Line 133. The authors mentioned that “The high f of 2BOICz, reminiscent of locally excited (LE) states”. What is the “reminiscent of locally excited (LE) states”? The meaning of the phrase is not clear.

Response: Thank you for your valuable comments.

In the manuscript, we have changed the sentence to “The high f value of 2BOICz is even comparable with or larger than that of the conventional fluorophors with locally excited (LE) states and should lead to both fast radiative decay and RISC process combining with its small ΔE_{ST} , which is exactly the aim of this molecular design.”

6. Caption of Figure S5. “The HOMO levels were determined to be -5.60 eV, -5.98 eV, and -6.02 eV for 1TICz, 1BOICz, and 2BOICz.” Please provide the method for obtaining HOMO levels via the ultraviolet photoelectron spectra (UPS). Why 1TICz shows shallower HOMO and LUMO compared to the other two molecules?

Response: Thank you for your valuable comments.

We have provided the method for obtaining HOMO levels via the ultraviolet photoelectron spectra (UPS) in the SI as shown in **Supplementary Fig.6**.

According to the molecular orbital theory, the energy levels of the molecule are determined by both donor and acceptor. A strong acceptor unit will deepen not only LUMO but also HOMO levels. The figure below compares the calculated energy levels of the donor and acceptor units, and the calculated molecular surface electrostatic potentials (ESP) of three compounds in their S_0 states. The oxygen-bridged boron acceptor (CF_3 -BO) unit displayed a deeper LUMO level around -2.28 eV than that of Trz (-1.80 eV), suggesting a stronger electron-accepting ability. Therefore, the introduction of acceptors with different push-pull electronic properties also can influence the ESP distribution to vary degrees. As shown in the figure below, the calculation results implied that the ESP distributions of 1TICz are more negative both in donor and acceptor groups than those of 1BOICz and 2BOICz. This is the reason that 1TICz shows shallower HOMO and LUMO compared to the other two compounds.

The work function of **1TICz**: $WF=21.22-16.67=4.55$ eV, the injection barrier relative to the substrate is **1.01** eV;
 The work function of **1BOCz**: $WF=21.22-16.22=5.00$ eV, the injection barrier relative to the substrate is **0.89** eV;
 The work function of **2BOCz**: $WF=21.22-16.26=4.96$ eV, the injection barrier relative to the substrate is **1.06** eV;
 The ionization potential of **1TICz**: $IP=21.22-(16.67-1.01)=5.56$ eV
 The ionization potential of **1BOCz**: $IP=21.22-(16.22-0.89)=5.89$ eV
 The ionization potential of **2BOCz**: $IP=21.22-(16.26-1.06)=6.02$ eV
 The above values of the ionization potential can be understood as the HOMO relative to the vacuum energy level.

Supplementary Fig. 6 The ultraviolet photoelectron spectra of the vacuum-deposited thin films for 1TICz, 1BOICz and 2BOICz. The HOMO levels were determined to be -5.56 eV, -5.89 eV and -6.02 eV for 1TICz, 1BOICz and 2BOICz.

Fig. The calculated LUMO levels of acceptor units. (Gaussian 09, B3LYP/6-31G (d)).

Fig. The molecular surface electrostatic potentials (ESP) of (a) 1TICz, (b) 1BOICz, and (c) 2BOICz in S_0 states (measuring scale (eV), red and blue indicate negative and positive electrostatic potentials, respectively).

7. The authors mentioned that the D-MR-A type TADF molecules 1BOICz and 2BOICz achieved smaller ΔE_{STs} and larger f values than D- π -A type TADF molecule 1TICz. However, Table S2 shows the PLQYs of 1TICz, 1BOICz, and 2BOICz (90%, 90%, and 89%, respectively). They are very close and do not correlate to the calculated f values. What's the sample condition used for the PLQY measurement? Please provide more discussion.

Response: Thank you for your valuable comments.

The absolute PLQY values of 1TICz, 1BOICz, and 2BOICz were measured in degassed toluene solution at a concentration of 10^{-5} M under 298 K. It is true that similar PLQYs were observed. In theory, the PLQY value is determined by both radiative and nonradiative transitions. Therefore, a large f value benefits an efficient radiative decay but does not

guarantee a high PLQY. It is speculated that the PLQYs in solution were influenced by the rotation of acceptors as only moderate dihedral angles between donor and acceptor. For 2BOICz, its two acceptors may induce more rotations for nonradiative transitions. We have also provided the PLQYs in the doped films and after suppressing the nonradiative transitions in solid films, all PLQYs of those emitters were improved and nearly ~100% was obtained for 2BOICz, in agreement with its highest f values.

8. The EQEs of the reported devices exceeded 40% when using 2BOICz. How many devices were fabricated and how was the standard deviation of EQE? Authors should carefully check the reproducibility.

Response: Thank you for your valuable comments.

To confirm the reproducibility of the peak EQE of the device based on 2BOICz, ten groups of devices (40 testing points) based on 20 wt% doped 2BOICz were fabricated in parallel. The statistics of the device EQEs show that the device performance is in good reproducibility and has been added in **Supplementary Fig. 17**.

Supplementary Fig. 17 The statistics of the peak EQEs based on 20 wt% doped 2BOICz measured from 40 devices, show an average peak EQE of 39.23% with a relative standard deviation of 1.62%.

9. Moreover, the ratio of horizontal emitting dipole orientation ($\Theta//$) of 2BOICz was 88% and its PLQY was only 89%. I am not sure the current data can support such a high EQE (41%) via the 2BOICz-based device. Please provide the reason and simulation to support it.

Response: Thank you for your valuable comments.

In fact, the device efficiency is related to the PLQY in solid film. The PLQY (89%) you mentioned was measured in the solution. While the PLQY value of 2BOICz was nearly unit (~100%) measured at a doping concentration of 20 wt% in mCPBC host matrix (shown in **Table 1**). As for the $\Theta//$ measurement, in our previous manuscript, we prepared the film samples and mailed them to Prof. Ma's group at South China University of Technology for characterization. We believed that the water vapor and oxygen probably damage the films during the transportation process. In the revised manuscript, as our group purchased the instruments now, we re-characterized the angle-dependent p-polarized PL spectra of three

compounds in 10 nm-thickness doped films (20 wt% emitters in mCPBC host matrix). As shown in the figure below, the high $\Theta_{//}$ values of 84%, 86% and 92% were obtained for 1TICz, 1BOICz and 2BOICz, respectively.

Actually, previously reported works with similar $\Theta_{//}$ and PLQY values have validated the potential to achieve remarkable $EQE \sim 40\%$ as illustrated in **Supplementary Table 7**. It is interesting to note that, since most reported TADF emitters based on solely LR-CT or solely SR-CT or hybrid LR-CT/LE excitations, the devices based on them showed relatively larger efficiency roll-off compared with our emitters, though their similar EQE_{max} . This further evidenced that our emitters with hybrid LR/SR-CT excitations could achieve a short-delayed component and thus suppress efficiency roll-off. To validate the high EQE values, we have also measured the angle-dependent EL intensities of the device based on 2BOICz and a nearly Lambertian profile pattern with a Lambertian coefficient of 0.98 was recorded. The EQE_{max} of 2BOICz device was re-calculated by Lambertian calibration to be still up to 40.4%, indicating that the high device efficiency was not overestimated.

Fig. (a) Angle-dependent PL spectra of 2BOICz, 1BOICz and 1TICz doped in mCPBC host. The experimental data are compared with the fitting curve for different horizontal dipole ratios. (b) Angle-dependent EL intensity of the TADF devices with 2BOICz, compared to the Lambertian distribution. The correction coefficient is calculated to be 0.981.

Supplementary Table 7 Summary of the reported TADF emitters with nearly 40% EQE_{max} values in OLED.

TADF materials	Refs	Φ_{PL} (%)	$\Theta_{//}$ (%)	λ_{EL} (nm)	$EQE_{max,100,1000}$ (%)
 2BOICz	This work	100%	92	528	41.2/41.1/36.6
 m-DPAcP-BNCz	[1]	97	86	496	42.0/36.7/17.5

 m-SF-BNCz	[1]	98	87	496	41.1/37.0/17.9
 BN-CPI	[2]	98	93	496	40.0/34.0/18.5
 t-Bu-v-DABNA HF	[3]	93	92	474	40.7/-/35.8
 TDBA-DI	[4]	99	89	-	38.15/-/34.03
 HF BN3	[5]	-	99	558	40.5/-/32.4
 DQBC	[6]	95	92	534	39.1/36.1/29.1
 CzDBA	[7]	100	84	528	38.0/-/37.9

Reference:

- [1] Q. Wang, Y. Xu, T. Yang, J. Xue, Y. Wang, *Adv. Mater.* 2022, 2205166.
- [2] P. Jiang, J. Miao, X. Cao, H. Xia, K. Pan, T. Hua, X. Lv, Z. Huang, Y. Zou, C. Yang, *Adv. Mater.* 2021, e2106954.
- [3] K. R. Naveen, H. Lee, R. Braveenth, D. Karthik, K. J. Yang, S. J. Hwang, J. H. Kwon, *Adv. Funct. Mater.* 2021, 32, 2110356.
- [4] D. H. Ahn, S. W. Kim, H. Lee, I. J. Ko, D. Karthik, J. Y. Lee, J. H. Kwon, *Nat. Photonics* 2019, 13, 540.

- [5] Y. X. Hu, J. Miao, T. Hua, Z. Huang, Y. Qi, Y. Zou, Y. Qiu, H. Xia, H. Liu, X. Cao, C. Yang, *Nat. Photonics* 2022, 16, 803.
- [6] Y. Chen, D. Zhang, Y. Zhang, X. Zeng, T. Huang, Z. Liu, G. Li, L. Duan, *Adv. Mater.* 2021, 33, e2103293.
- [7] T.-L. Wu, M.-J. Huang, C.-C. Lin, P.-Y. Huang, T.-Y. Chou, R.-W. Chen-Cheng, H.-W. Lin, R.-S. Liu, C.-H. Cheng, *Nat. Photonics* 2018, 12, 235.

Besides a high PLQY and $\Theta_{//}$, the device light out-coupling efficiency (η_{out}) is also greatly influenced by the refractive index of the hole transporting layer. In our devices, we adopted TAPC as the hole-transporting layer (HTL), which is well known by its low refractive index (1.67) to enhance the η_{out} . Actually, we have compared the device performances with TAPC and NPB (a regular refractive index of 1.76) as the HTL and the device performances were illustrated in **Supplementary Fig. 19**. Clearly, the device with TAPC showed an improved EQE_{max} compared with that of NPB, evidencing that a low refractive index HTL layer also contributes to the high EQE_{max} of the device.

Supplementary Fig. 19 The effect of the hole transport layer (TAPC and NPB) on the device performance. The device structure: ITO/ TAPC or NPB (40 nm)/ TCTA (10 nm)/ mCPBC: 20 wt% 2BOICz (24 nm)/ CzPhPy (10 nm)/ DPPyA (30 nm)/ LiF (0.5 nm)/ Al (150 nm). (a) Luminance-voltage-current density characteristics. (b) External quantum efficiency (EQE), (c) current efficiency (CE) and (d) power efficiency (PE) versus luminance.

Revised Manuscript:

We have also measured the angle-dependent EL intensities of the device based on 2BOICz and a nearly Lambertian profile pattern with a Lambertian coefficient of 0.98 was recorded as shown in Supplementary Fig. 17a. The EQE_{max} of 2BOICz device was re-calculated by

Lambertian calibration to be still up to 40.4%, indicating that the high device efficiency was not overestimated. To confirm the reproducibility of the peak EQE_{max} of the device based on 2BOICz, ten groups of devices (40 testing points) were fabricated in parallel. The statistics of those EQE_{max} s show that the device performance is in good reproducibility (Supplementary Fig. 17b).

.....

Besides, the low refractive index of the hole-transporting material, TAPC, is also beneficial to this high EQE. As a comparison, we have also fabricated a device with another most commonly used hole-transporting material, NPB, which possesses a relatively higher refractive index, and a relatively lower EQE_{max} of 39% was realized (Supplementary Fig. 19). Therefore, the excellent TADF properties, a high $\Theta_{//}$ and an optimized device structure should concurrently account for the cutting-edge EQE_{max} of 2BOICz. In fact, previous works have also demonstrated EQE_{max} s of over 40% using TADF emitters with similar Φ and $\Theta_{//}$ values as illustrated in Supplementary Table 7. It is interesting to note that most of those previously reported emitters showed relatively larger efficiency roll-off compared with 2BOICz, though their similar EQE_{max} s. This further evidenced the advantages of our emitters with hybrid LR/SR-CT excitations to achieve a short-delayed component for an alleviated efficiency roll-off.

10. Despite the relatively large ΔE_{ST} (0.17 eV) of 1TICz, the delayed lifetime value is only 3222 ns. Do T_n states of 1TICz participate in the RISC process? In addition, the phosphorescence pattern of 1TICz looks dissimilar (Fig 2a) among the three compounds, how to make sure whether it originates from the CT character?

Response: Thank you for your valuable comments.

We noticed that the time scale of the measurement window for the time-resolved PL transient decay curve of 1TICz was not long enough and we have carefully re-conducted this measurement and recorded a relatively longer delayed component of 9889 ns. The corresponding rate constants have been updated in **Table 1**.

To confirm whether T_n states of 1TICz participate in the RISC process, the spin-orbit couplings (SOC) between S_1 and T_n states were calculated using a TD-DFT approach at the B3LYP/6-31G(d,p) level. Interestingly, the T_2 state of 1TICz indeed participates in the RISC process with a high $\langle S_1 | \hat{H}_{\text{soc}} | T_2 \rangle$ value of 0.13 cm^{-1} .

The exact reason for the dissimilar phosphorescence pattern of 1TICz among the three compounds is unclear. Phosphorescence with a slight vibronic spectrum from CT states has also been observed in literature and the reason was also unclear (Adachi et al, *Sci. Adv.* **2018**, *4*, eaao6910). To validate its CT character, we measured the phosphorescence in hosts with different polarities. As illustrated in Figure below, a significant redshifted was observed for the phosphorescence measured in high polarity host PPF (ground-state dipole moment, $\mu_{\text{GS}} = 5.8 \text{ D}$) and DPEPO ($\mu_{\text{GS}} = 7.9 \text{ D}$) compared with the low polarity host mCPBC ($\mu_{\text{GS}} = 3.63 \text{ D}$). It is the CT states that can be stabilized in a more polarity matrix. Therefore, we concluded that the phosphorescence of 1TICz originates from the CT character.

Fig 2d Transient PL decay curves of 20 wt% 1TICz doped films.

Fig. The measured phosphorescence spectra of 20 wt% 1TICz doped in the different hosts (mCPBC, PPF and DPEPO) at 77 K.

11. In Table S3 to Table S5, as the concentration of 2BOICz-based devices increase from 20 wt% to 40 wt%, the EQE_{max} slightly decrease from 41.2% to 38.9%. However, why the EL peak wavelengths of 2BOICz are sensitive to the concentrations? It shows a serious red-shifted peak and reveals a high degree of aggregation that may occur when increasing the doping concentration. Please explain.

Response: Thank you for your valuable comments.

The significant red-shifted emission peaks of 2BOICz should be attributed to the increased polarity of the doped films with increasing dopant concentrations. It has been widely recognized that the emission profile of the TADF emitters with charge transfer transitions can be stabilized by the high polarity of the environment in both solvent and solid films. In the host-dopant systems, the polarity of the doped films can also be enlarged by increasing the concentrations of dopant with strong CT-type transitions, which is known as ‘self-solid-state solvation’ (Ref: H.S. Kim et al., *J. Phys. Chem. C*, **2017**, *121*, 13986). Given the strong electron withdrawing ability of acceptors, 2BOICz itself would show high polarity in solid films and thus induce significant redshifted emission. Actually, in our devices, more obvious red-shifted emissions were observed for the 1BOICz and 2BOICz based devices compared

with that of 1TICz. This can be well explained by the much larger excited-state dipole moments (μ_{ES}) of 1BOICz (1.11 D) and 2BOICz (1.16) than 1TICz (0.37 D).

We added the sentence to discuss this point in Device characterization and performance section of the manuscript: “**Supplementary Fig. 11-16** revealed an interesting phenomenon that both 1BOICz and 2BOICz based devices exhibited obviously red-shifted emissions with increased dopant concentrations while the one with 1TICz only showing limited red-shift. This can be well explained by the much larger excited-state dipole moments of 1BOICz (1.11 D) and 2BOICz (1.16 D) than 1TICz (0.37 D), induced by the strong electron-withdrawing ability of CF₃ substituted BO-acceptors.”

Reviewer #2 (Remarks to the Author):

The paper deals with new TADF emitters using multiresonance bridge. The new emitters showed a large decay rate over an order of magnitude higher than ISC rate with short-delayed lifetime. However, the B-O accept was already many reported with high efficient TADF materials. Authors should be compared with IBOICz and 2BOICz without electron acceptor CF₃ groups at least in the calculation. The other groups already used {A with MR-D concept}. Therefore, the author should be explained the author's [A-MR-D concept] is different with the already reported [A with MR-D concept]. Although the large decay rate with short-delayed life time, FWHM of IBOICz and 2BOICz showed still broad without MR effect. I did not find the scientific improvement at these points. I do not recommend to publish in the Nature Communication.

Ref) Ahn, D.H. et al. Nat. Photonics 13, 540-546 (2019)

Response: Thank you for your valuable comments.

We are sorry for the misunderstanding induced by the “D-MR-A” abbreviation. In the revised manuscript, the description of the emitter has been changed to “TADF emitters with hybrid LR/SR-CT excitations”, namely TADF emitter with hybridized long-range (LR) and short-range (SR) charge transfer (CT) transitions. Furthermore, to better understand and strengthen the merits of this manuscript, we have changed the title to ‘*High-efficiency and stable short-delayed fluorescence emitters with hybrid long- and short-range charge-transfer excitations*’. We have been provided extensive revisions including in the Abstract and Introduction section to discuss the design principle for TADF emitters with different electronic excitations.

Previously reported D-A type TADF emitters always bear solely LR-CT or hybrid LR-CT/localized excited (LE) excitations, which face a mutual exclusion of a small singlet-triplet energy gap (ΔE_{ST}) and a large oscillator strength (f). This trade-off has been widely recognized as the main reason for a long-lifetime delayed component and will induce a significant efficiency roll-off and poor operation stability under high luminance. The aim of our work here is to propose a new paradigm of TADF emitters with hybrid LR/SR-CT excitations to break the above trade-off to balance a small ΔE_{ST} and a large f for a short-delayed component. As commented by Reviewer 1, our work here provided “the novel molecular design and outstanding device efficiency”.

Though their relatively wide spectrum, D-A type TADF emitters are still highly desired as their potential as not only emitters for lighting applications but also sensitizers for fluorescence emitters, known as hyper-fluorescence or TADF-sensitizer fluorescence (TSF). Particularly, the combination of a TADF sensitizer and a multi-resonance (MR) emitter has been a common strategy to avoid the inefficient RISC issue of MR emitter for high efficiency, narrowband emission and long device lifetime (*Nat. Photonics* **2021**, *15*, 203; *Nat. Photonics* **2021**, *15*, 208). An ideal TADF molecule as either emitter or sensitizer should possess a short-lifetime delayed component for a fast and efficient exciton consumption to guarantee good device performances under practical high luminance. To achieve a short-delayed lifetime, a small ΔE_{ST} for an efficient RISC process and a large f value for efficient radiative decay should be satisfied simultaneously. However, in theory, a small ΔE_{ST} requires a spatially separated distribution of HOMO and LUMO while a large f requires an efficient overlap of HOMO and LUMO, which are principally conflicting with each other. The orbital distribution

therefore determines the performances of molecules. Commonly, a long-range charge transfer (LR-CT) type orbital distribution means a large separation of HOMO and LUMO distributions, thus favoring a small ΔE_{ST} but also a small f value. The localized excited (LE) type orbital distribution, on the contrary, means an efficient HOMO-LUMO overlap, thus benefiting a large f value but also a large ΔE_{ST} . Nearly all D-A type TADF compounds in previous works can be divided into two categories as illustrated in the figure below, one is LR-CT type and another one is LR-CT/LE type. Considering that the LR-CT type would always favor a small ΔE_{ST} but also a small f , researchers developed TADF emitters with hybrid LR-CT and LE (LR-CT/LE) excitations, aiming to achieve a balance by controlling the ratio of LR-CT and LE orbitals. However, it is rather difficult and though some efficient TADF emitters have been developed, the progress is still tardy as most TADF emitters still suffer from a long-lifetime delayed component.

Recently, a new TADF emitter, that is the multi-resonance emitter, has emerged, bearing a new type of short-range CT (SR-CT) orbital distribution. This SR-CT excitation facilitates a similar or even larger f value and also a relatively smaller ΔE_{ST} compared with the LE type, but the ΔE_{ST} is not small enough to trigger an efficient RISC process compared with the LR-CT type. Besides, according to the Marcus-Levich-Jortner theory, the small Huang-Rhys factors of those molecules also render them rather slow RISC rates (k_{RISC}) in the range of 10^4 s⁻¹ and even milli-second-scale τ_D (*Nat. Commun.* **2022**, *13*, 4876).

Inspired by the LR-CT/LE type emitters and considering the characters of those excitation types, we envisioned that hybridizing an auxiliary SR-CT with the main LR-CT excitation may not only avoid their disadvantages but also combine their complementary advantages to afford a new paradigm of TADF molecules simultaneously with a small ΔE_{ST} for a fast RISC process and a large f for an efficient radiative decay. With this in mind, we demonstrated a strategic implementation of TADF emitters with hybrid LR/SR-CT excitations by incorporating MR-acceptor groups on to a sterically uncrowded donor segment. This architecture, on one hand, created a hybrid electronic excitation with a main donor-to-acceptor LR-CT and an auxiliary bridge-phenyl SR-CT characteristics, balancing a small ΔE_{ST} and a large f . On the other hand, the adoption of equivalent dual multi-resonance acceptors was unveiled to double f without influencing ΔE_{ST} and also enlarged the horizontal dipole ratio for enhanced light extraction. The two proof-of-the-concept molecules successfully realized short-lifetime delayed components and unprecedented high EQEs with low roll-off and extended device lifetime. Our achievement here shatters the stereotypical physicochemical views of TADF emitters limited by the mutual exclusion of a small ΔE_{ST} and a large f , potentially revolutionizing the molecular design principle and unlocking the full potential of TADF OLEDs.

Fig. The design principle for TADF emitters with different electronic excitations.

Though some TADF emitters based on B-O acceptors have been developed, our molecular design concept is totally different from previous works (more details are discussed in **Part A**). All B-O based TADF emitters in previous works inherited LR-CT or LR-CT/LE type orbital distributions. Our emitters here bear a new kind of hybrid LR/SR-CT excitations. Besides, we first introduced CF₃ groups at the *para*-positions of the B atom to enhance the electron withdrawing ability and enhance the SR-CT on the attached phenyl rings. More importantly, we also demonstrated a dual acceptor structure can further enlarge the *f* value without influencing ΔE_{ST} , which provided a screen that one can construct TADF emitters with equivalent multiple acceptors (or donors) to generate equivalent multiple CT channels to multiply the *f* value without influencing the small ΔE_{ST} . Interestingly, during the revision of our manuscript, we noticed that Prof. Kwon using a BO acceptor with two donors also benefits a faster *k*_{RISC} (*Nat. Commun.* **2023**, *14*, 419). But still, only LR-CT type orbital transition was observed for their emitters (**Supplementary Table 8**).

In addition, our molecular design concept is also clearly different with the previous A with MR-D concept (more details are discussed in **Part B**), which, actually was also first reported by our group (*Angew. Chem. Int. Ed.* **2019**, *58*, 16912; *Angew. Chem. Int. Ed.* **2020**, *59*, 17499). The A with MR-D structure was proposed mainly aiming at realizing a red-shift the emission color and only slow RISC processes were obtained. Furthermore, in the revised manuscript, we have also constructed narrowband emitting devices using our molecules as sensitizers for a MR emitter, achieving not only high maximum EQE but also smaller efficiency roll-off compared with the device using the reference TADF compound as sensitizer (more details are discussed in **Part C**). Those results demonstrated that our molecular design strategy can generate not only ideal TADF emitters but also ideal TADF sensitizers for MR emitters, which will arouse wide attentions.

We can explain our idea in detail based on your comments as follows:

Part A: “However, the B-O accept was already many reported with high efficient TADF materials. Authors should be compared with IBOICz and 2BOICz without electron acceptor CF₃ groups at least in the calculation”

Response: It is true that D-A or A-D-A/D-A-D type TADF materials based on BO acceptors have been reported as summaries in **Supplementary Table 8**, represented by the one reported by Kwon et al (*Nat. Photonics* **2019**, *13*, 540). Of particular note is that all those TADF emitters mainly adopted large sterically crowded donor groups, resulting in highly twisted geometries with large dihedral angles ($>70^\circ$). Therefore, nearly all those compounds showed totally separated HOMO-LUMO distributions for solely LR-CT excitations, which favor a small ΔE_{ST} but not a large *f* value.

Supplementary Table 8 Summary HOMO/LUMO orbital distributions, ΔE_{ST} and f values of this work and the previously reported D-A or A-D-A/D-A-D type TADF emitters based on the oxygen-bridged boron acceptor motif.

TADF materials	Refs	HOMO	LUMO	ΔE_{ST} (eV)	f
 2BOICz	This work			0.10	0.35
 TDBA-Ac	Nat. Photonics. 2019, 13, 540			0.00 6	0.00 01
 TDBA-DI	Nat. Photonics. 2019, 13, 540			0.07 3	0.03 1
 TMCz-BO	Nat. Commun. 2020, 11, 1765			0.00 6	0.00 06
 TMCz-3P	Nat. Commun. 2020, 11, 1765			0.00 5	0.00 05
 pMDBA-DI	Adv. Funct. Mater. 2021, 32, 2110356			0.07 8	0.02 7
 mMDBA-DI	Adv. Funct. Mater. 2021, 32, 2110356			0.09 7	0.03 3

 DBA-BFICz	Adv. Funct. Mater. 2021, 31, 2105805			0.08 4	0.05 6
 DBA-BTICz	Adv. Funct. Mater. 2021, 31, 2105805			0.11 1	0.03 3
 PzTDBA	Adv. Mater. 2021, 33, 2007724			0.00 5	0
 PzDBA	Adv. Mater. 2021, 33, 2007724			0.00 5	0
 DBA-DmICz	Nat. Commun. 2023, 14, 419			0.02 3	0.04 6
 DBA-DTMCz	Nat. Commun. 2023, 14, 419			0.00 8	0

We also chose TDBA-DI as a representative molecule, and the comparison between it with the investigated molecules is shown in figure below. It is worth noting that the optimized TDBA-DI showed a large dihedral angle (θ) of 71.6° and thus no HOMO distribution extended to the attached phenyl rings. Consequently, only LR-CT excitation was observed, which accounts for a rather small ΔE_{ST} of 0.0701 eV and also a small f of 0.0309. For 1TICz compound, being another type of TADF emitter, showed hybrid LR-CT and LE (LR-CT/LE) orbital distributions. A much larger f value of 0.2361 was obtained but also a large ΔE_{ST} of 0.1934. While for our molecule, 1BOICz, a clear hybrid LR-CT and SR-CT (LR/SR-CT) orbital distribution was observed. And a small ΔE_{ST} of 0.1054 eV and a large f of 0.1462 was observed, realizing a balance. Clearly, our molecular design here provided an alternative molecular distribution situation, which theoretically and experimentally benefits breaking the dilemma between a small ΔE_{ST} and a large f to finally obtain a short-lifetime delayed component.

Fig. The design principle for conventional TADF emitters (LR-CT or LR-CT/LE type orbital distributions), and this work proposed hybrid LR-CT and SR-CT (LR/SR-CT) orbital distribution. The optimized structures of typical TADF emitter (TDBA-DI) and the two compounds investigated in this work. The calculated HOMO-LUMO orbital distribution, related values of the dihedral angle (θ), singlet-triplet energy gap (ΔE_{ST}), and oscillator strength (f).

According to your suggestions, we have also compared the molecules with and without CF_3 groups. The adoption of CF_3 groups was aimed at, on one hand, enhancing the electron-withdrawing ability of the BO-acceptors to avoid the influence of the low-lying localized excited (LE) triplet states of the donor, on the other hand, enhancing the MR distributions on the attached phenyl rings for more balanced f and ΔE_{ST} values. It can be clarified as follows.

Firstly, the adoption of CF_3 groups can enhance the electron-withdrawing ability of the BO acceptor. In our work, 5,11-dihydroindolo[3,2-b]carbazole (32bICz) group was adopted as the donor, which possessed a LE triplet state of only 2.65 eV. Therefore, to avoid the influence of LE states on ΔE_{ST} , a strong electron-withdrawing unit should be adopted. We calculated the LUMO levels of the CF_3 -BO unit in our manuscript and other several BO derivatives in previous reports. As illustrated in the figure below, CF_3 -BO showed the deepest LUMO energy, suggesting its strongest electron-withdrawing ability. Using strong acceptors, the energy of the charge transfer (CT) states can be greatly reduced, facilitating to avoid the influence of the low-lying LE states.

Fig. Chemical structures, the contrast of electron-withdrawing ability of DBNA based acceptor derivatives by decorating peripheral groups. LUMO levels were calculated by Gaussian 16, B3LYP/6-31G (d).

To demonstrate our idea, we have synthesized a reference compound (2BOICz-tBu) with 2,12-di-tert-butyl-5,9-dioxa-13b-boranaphtho[3,2,1-de]anthracene (tBu-BO) as the acceptor. Both compounds showed a wide structureless fluorescence emission, evidencing the CT characteristics of their singlet states. Owing to its weak acceptors, 2BOICz-tBu showed a significantly blue-shifted emission with a higher S_1 energy of 2.90 eV than that of 2.68 eV for 2BOICz (**Supplementary Fig. 8**). Moreover, clear differences were also observed from their phosphorescence. For 2BOICz, its phosphorescence spectrum also showed a clear CT character and a small ΔE_{ST} of 0.06 eV was obtained for an efficient RISC process. On the contrary, 2BOICz-tBu showed a well-resolved phosphorescence spectrum resembling that of the donor (ICz-Ph), suggesting its LE character of the triplet state (2.65 eV). A large ΔE_{ST} of 0.25 eV was thereafter obtained, thus leading to a poor RISC process.

On the other hand, the existence of CF_3 groups could also enhance the MR type orbital distributions on the attached phenyl rings. As illustrated in **Supplementary Fig. 8**, similar dihedral angles (θ) between donor and acceptors of 2BOICz-tBu were observed compared with that of 2BOICz. Interestingly, its frontier molecular orbitals (FMOs) also inherit a hybridized orbital distribution character similar to 2BOICz. But the difference can also be observed. For 2BOICz, both HOMO and LUMO distributions on the attached phenyl rings were well constrained on the single atoms and thus a small overlap integral (0.2950) for a small ΔE_{ST} (0.1002 eV). On the contrary, the LUMO distribution on the bridge phenyl rings of 2BOICz-tBu extended to the adjacent atoms, thus enhancing orbital overlaps with a relatively larger overlap integral (0.3045) and thus a relatively larger ΔE_{ST} (0.1270 eV). Therefore, the existence of CF_3 substitutions can perform better in balancing a small ΔE_{ST} and a large f .

Supplementary Fig. 8 (a, b) The chemical structures of 2BOICz and 2BOICz-tBu, and the calculated values of the dihedral angle (θ), HOMO-LUMO distributions, singlet energy level (S_1) and oscillator strength (f). Furthermore, the overlap integral of the orbitals at S_1 state is calculated by Natural Transition Orbital (NTO) analysis. (c) UV-vis absorption (left axis) in toluene (10^{-5} M), normalized PL (298 K) and phosphorescence (77 K) spectra (right axis) of 2BOICz-tBu in toluene. (d, e) Fluorescence (298 K or 77 K) and phosphorescence (77 K) spectra of doped films (20 wt% doped films).

Based on those findings, we found that several criteria should be satisfied to maximize the performances of TADF emitters with LR/SR-CT excitations. Firstly, a moderate dihedral angle between donor and MR-acceptor segments is necessary to guarantee HOMO extension to the bridge phenyl rings. Secondly, the LE triplet states of donor and MR-acceptor should be energetically close to or higher than CT states to avoid its influence on ΔE_{ST} . Thirdly, proper substitutions on MR-acceptors should be considered to enhance the MR characters to guarantee the SR-CT excitations on the bridge phenyl rings.

In the Supporting Information of the revised manuscript, the detailed study on the properties of 2BOICz-tBu has been provided as a contrast. The synthetic procedure and related characterizations were presented in the Experimental section and Supplementary Figure section. The UV-vis absorption and emission spectra in toluene (10^{-5} M) were shown in **Supplementary Fig. 9a**, the relatively strong absorption strength was observed in the long wavelength, accompanied with a bright deep-blue emission ($\lambda_{PL}=441 \text{ nm}$, $\Phi_{PL}=85\%$), this result matches well with the calculation that showed a high f value. The solvatochromic effect of 2BOICz-tBu in various solvents further confirmed the strong CT feature in its S_1 state (**Supplementary Fig. 9b**). Furthermore, the 20 wt% 2BOICz-tBu doped in PPF film displayed blue emission with a peak at 466 nm, and an excellent Φ_{PL} of 88%. Especially, 2BOICz-tBu revealed a higher ΔE_{ST} of 0.25 eV due to its lower triplet energy compared with that of 2BOICz (ΔE_{ST} of 0.06 eV). Due to the relatively large ΔE_{ST} , 2BOICz-tBu exhibited a negligible TADF behavior in the doped film, only showed a prompt decay lifetime in the

order of nanoseconds (**Supplementary Fig. 9c**).

To evaluate the EL performance of 2BOICz-tBu, we fabricated the OLED devices whose structure is displayed in **Supplementary Fig. 20a**. PPF was used as the host and hole-blocking layer due to its high T_1 (3.1 eV) energy level and shallow HOMO level. As shown in **Supplementary Fig. 20b**, the 20 wt%-40 wt% doped devices exhibited blue EL emission with peaks at ~465 nm and FWHMs of ~56 nm. The current density-voltage-luminance (J-V-L), and EQE, PE, CE versus luminance curves are plotted in **Supplementary Fig. 20c-20f**, and the EL performance parameters were summarized in **Supplementary Table 6**. The devices showed turn-on voltages in the range of 2.9-3.3 V and obtained the highest EQE_{max} value of 17.7%. The EL decay curves of those devices showed long-lasting delayed components, suggesting the TADF characteristic of this emitter though being weak, which accounts for this moderate EQE_{max} . Albeit the relatively large ΔE_{ST} , the RISC process should arise from the participation of the high-lying triplet excited state (T_n) (DFT calculation section). The Θ_{\parallel} factor of 2BOICz-tBu doped PPF film was also analyzed as shown in **Supplementary Fig. 20h**. The experimental result confirmed a high Θ_{\parallel} factor of 89.7% for 2BOICz-tBu emitter, which is also beneficial to enhance the outcoupling efficiency. However, it should be pointed out that, 2BOICz-tBu based device showed not only a much lower EQE_{max} but also a significant efficiency roll-off compared with the device using 2BOICz. All those results provide insight that the CF_3 units on the *para*-position of the BO motif play a critical role in the modulation of excited energies and optoelectronic properties of the investigated compounds in this work.

The synthetic procedure of 2BOICz-tBu.

¹H NMR spectrum of 2BOICz-tBu in CD₂Cl₂.

Mass spectrometry of 2BOICz-tBu.

Supplementary Fig. 7 (a) The ultraviolet photoelectron spectra of the vacuum-deposited thin films for 2BOICz-tBu. The HOMO levels were determined to be -5.50 eV. The band energy is 2.98 eV, the LUMO energy level can be estimated to be -2.52 eV. (b) Decomposition temperature (T_d) with 5% weight loss for 2BOICz-tBu.

Supplementary Fig. 9 (a) UV-vis absorption (left axis) in toluene (10^{-5} M), and normalized PL spectra (right axis) of 2BOICz-tBu in toluene. (b) The solvent-dependent PL spectra. (c) Transient PL decay curve of 20 wt% 2BOICz-tBu doped film. (d) The spin-orbit coupling (SOC) matrix elements between S_1 and T_n states of 2BOICz-tBu were conducted at the B3LYP/6-31G(d,p) level.

Supplementary Fig. 20 (a) Device architecture for 2BOICz-tBu based TADF-OLEDs. (b) EL spectra of the devices under 1000 cd m⁻². (c) Luminance-voltage-current density characteristics. (d) External quantum efficiency (EQE), (e) current efficiency (CE) and (f) power efficiency (PE) versus luminance. (g) EL transient spectra for the 20 wt%, 30 wt% and

40 wt% 2BOICz-tBu doped devices. (h) Angle-dependent PL spectra of 20 wt% 2BOICz-tBu in PPF host.

Supplementary Table 6 Summary of the EL performance of 2BOICz-tBu based devices.

Doping	$\lambda_{\text{EL}}^{\text{a}}$	V_{on}^{b}	$L_{\text{max}}^{\text{c}}$	$\text{CE}_{\text{max}/100}^{\text{d}}$	$\text{PE}_{\text{max}/100}^{\text{e}}$	$\text{EQE}_{\text{max}/100}^{\text{f}}$	CIE
concentration [nm]	[V]	$[\text{cd m}^{-2}]$	$[\text{cd A}^{-1}]$	$[\text{lm W}^{-1}]$	[%]	(x,y) ^g	
20 wt%	467	3.3	3987	27.5/15.6	27.0/10.2	17.7/10.1	(0.157, 0.201)
30 wt%	468	3.0	5200	21.0/15.3	22.0/10.0	15.1/9.8	(0.155, 0.214)
40 wt%	469	2.9	6589	18.8/15.9	18.5/10.8	12.5/9.8	(0.156, 0.212)

^a EL peak wavelength; ^b Turn-on voltage (V_{on}); ^c Maximum luminescence (L); ^d Maximum current efficiency (CE), value at 100 cd cm^{-2} ; ^e Maximum power efficiency (PE), value at 100 cd cm^{-2} ; ^f Maximum external quantum efficiency (EQE), value at 100 cd cm^{-2} ; ^g CIE coordinates at 1000 cd cm^{-2} .

Part B: “The other groups already used {A with MR-D concept}. Therefore, the author should be explained the author’s [A-MR-D concept] is different with the already reported [A with MR-D concept].”

Response: Thank you for your valuable comments.

The concept of A with MR-D was actually first reported by our group. This concept was mainly adopted to enhance the CT property of the MR emitter to red-shift the emission colors (*Angew. Chem. Int. Ed.* **2019**, *58*, 16912; *Angew. Chem. Int. Ed.* **2020**, *59*, 17499). In those compounds, the SR-CT is the main one and thus a large f value can be anticipated but also a large ΔE_{ST} . Consequently, only poor RISC rates in the range of 10^4 s^{-1} can be obtained with rather long lifetimes of delayed compounds. And the corresponding devices suffer from significant efficiency roll-off at practical high luminance.

On the contrary, our molecules here possessed a main LR-CT and an auxiliary SR-CT excitation, thus facilitating not only a small ΔE_{ST} but also a large f . Therefore, a fast triplet up-conversion and singlet radiation process was achieved concurrently in our molecules and the corresponding OLED devices demonstrate an impressive performance with alleviated efficiency roll-offs.

Part C: “Although the large decay rate with short-delayed lifetime, FWHM of IBOICz and 2BOICz showed still broad without MR effect.”

Response: We are sorry for the misunderstanding induced by the “D-MR-A” abbreviation. This description has been replaced by “TADF emitters with hybrid LR/SR-CT excitations”. The aim of our work here mainly focused on resolving the mutual exclusion of a small singlet-triplet energy gap (ΔE_{ST}) and a large oscillator strength (f) faced by D-A type TADF molecules. Though the relatively broad emission spectra, conventional D-A type TADF molecules are still highly desired. On one hand, broad emission spectra are highly desired for white OLEDs as solid-state lighting applications. On the other hand, TADF molecules can also be utilized as sensitizers for fluorescent emitters, known as hyperfluorescence (HF) or TADF-sensitized fluorescence (TSF). Particularly, the combination of a TADF sensitizer and

a MR final emitter has been a common strategy in literature, which resolves not only the issue of broad spectra of TADF molecules, but also the drawback of inefficient RISC process of MR compounds. Actually, based on such strategy, narrowband emission, high efficiency and long operation lifetime can be anticipated for OLEDs, which have been widely studied. A balanced k_f and k_{RISC} are highly desired for TADF molecules as not only emitters and also sensitizers. As an ideal sensitizer, a large k_{RISC} should benefit triplet harvesting and a large k_f favors to enhance the desired Förster energy transfer.

We therefore envisioned that besides as emitters, 1BOICz and 2BOICz should also perform well as sensitizer for MR emitters. To verify this, TSF devices were fabricated using 1BOICz and 2BOICz as sensitizers, and a previously reported yellow MR-TADF emitter, DBN-ICz, was adopted as the narrowband emitter (*Adv. Mater.* **2022**, e2209396). For comparison, 1TICz was also adopted as sensitizer. All three TADF molecules provided efficient spectral overlapping between their emission spectra with the absorption spectrum of DBN-ICz emitter (**Supplementary Fig. 21**), affording large Förster resonance energy transfer (FRET) radius (R_0) of 3.4, 3.3, and 3.3 nm for 1TICz, 1BOICz, and 2BOICz based systems, respectively, which imply the efficient energy transfer in these sensitizing systems. Furthermore, all three films of mCPBC: 20 wt% sensitizer: 1 wt% DBN-ICz exhibited the same spectra peaking at 551 nm with a narrow FWHM of ~25 nm and the Φ_{PL} values of 94%, 97% and 98 % for 1TICz, 1BOICz and 2BOICz doped films, respectively.

The TSF-OLEDs were constructed with all the other functional layers that were the same as the TADF devices in the manuscript except the emitting layers of mCPBC: 20 wt% sensitizer: 1 wt% DBN-ICz. The detailed device configuration is depicted in **Supplementary Fig. 22** as well as the corresponding plot of current density (J), and luminance (L) versus voltage (V), and power efficiency (PE), and current efficiency (CE) versus luminance. All three TSF devices displayed sharp spectra with peak at 551 nm, small FWHMs of 25 nm and CIE coordinates of (0.37, 0.61) (**Fig. 5**), representing pure orange emissions. Impressively high maximum external quantum efficiencies of 37.4% and 38.4% were obtained for the 1BOICz and 2BOICz based TSF devices, which is significantly higher than the one based on 1TICz (33.1%) as sensitizer. Moreover, the TSF devices based on 1BOICz and 2BOICz showed greatly alleviated efficiency roll-off, with EQE remaining 35.2% and 36.1% at 1,000 cd m^{-2} , respectively, being about 94% of their maximum values. On the contrary, an EQE of 27.0% was observed for 1TICz based device, being 81% of its maximum. The relatively higher efficiencies with smaller efficiency roll-off of devices based on 1BOICz and 2BOICz should be attributed to their more balanced k_f and k_{RISC} than those of 1TICz. The EL decay curves of those three TSF devices were also measured at 1,000 cd m^{-2} , showing obviously shorter delayed lifetimes of devices using 1BOICz and 2BOICz, which is similar to the results obtained from devices that used them directly as TADF emitters. This discrepancy can be attributed to the more balanced k_f and k_{RISC} of 1BOICz and 2BOICz, which not only facilitates an efficient T_1 up-conversion process but also favors fast energy transfer, thus quickening the exciton radiative consumption process to reduce the exciton accumulation under high current density. This finally leads to high device efficiency with low roll-off. The state-of-the-art performances validated the superiority of 1BOICz and 2BOICz as sensitizers and further evidenced the effectiveness of our molecular design.

Supplementary Fig. 21 (a) The chemical structure of narrowband yellow MR emitter DBN-ICz. (b, c, and d) UV-Vis absorption of DBN-ICz in toluene with a concentration of 10^{-5} mol^{-1} , the PL spectra of 20 wt% 1TICz (or 1BOICz, 2BOICz) and 20 wt% 1TICz (or 1BOICz, 2BOICz): 1 wt% DBN-ICz doped in mCPBC.

Fig. 5 TSF-OLED device performances. **a**, Normalized EL spectra under $1,000 \text{ cd m}^{-2}$. **b**, EQE versus luminance characteristics, and the narrowband yellow MR emitter DBN-ICz.

Supplementary Fig. 22 (a) Device architecture for TSF-OLEDs based 1TICz, 1BOICz and 2BOICz. (b) Luminance-voltage-current density characteristics. (c) The current efficiency (CE) and (d) power efficiency (PE) versus luminance. (e) EL transient spectra for the TSF-OLEDs. (f-h) Angle-dependent EL intensities of the TSF-OLEDs with 2BOICz, 1BOICz and 1TICz, compared to the Lambertian distribution, the correction coefficient is calculated to be 0.980, 0.972, 0.970, respectively.

Reviewer #3 (Remarks to the Author):

The authors have claimed that donor-multiresonance bridge-acceptor typed emitters can achieve a small singlet-triplet energy gap and a large oscillator strength. They demonstrated that the D-MR-A emitter showed high efficiency with low roll-off due to a fast RICS rate of over 10^6 /s. However, this reviewer is not positive about supporting the publication of this manuscript in Nature Commun due to the following concerns.

1) The substantial merit of the MR core in the TADF emitters is a narrow emission spectrum giving a high color purity. Indeed, the typical DABNA-based MR emitter has a very low RISC rate of below 10^4 /s. Therefore, reducing the singlet-triplet energy gap and enhancing the oscillator strength are key issues in improving the optoelectrical characteristics of OLEDs. In this concern, this reviewer could not find merit in the D-MR-A emitter claimed in this manuscript.

Response: Thank you for your valuable comment.

We are sorry for the misunderstanding induced by the “D-MR-A emitter” abbreviation. A more precise description has been provided in the manuscript as “TADF emitters with hybrid LR/SR-CT excitations”, that is TADF emitters with hybridized long-range (LR) and short-range (SR) charge transfer (CT) excitations. As you mentioned, “reducing the singlet-triplet energy gap and enhancing the oscillator strength are key issues in improving the optoelectrical characteristics of OLEDs.” Actually, not only MR emitters but also conventional D-A type TADF emitters still face the mutual exclusion of a small singlet-triplet energy gap (ΔE_{ST}) and a large oscillator strength (f). This trade-off has been recognized as the main reason for a long-lifetime delayed component and will induce a significant efficiency roll-off and poor stability under high luminance. Our molecules here aimed at breaking this trade-off to develop TADF molecules with a balanced small (ΔE_{ST}) and large f .

The orbital distributions essentially determine the performances of TADF emitters. Previously D-A or D-A-D/A-D-A type TADF emitters always inherited LR-CT or hybrid LR-CT and localized excited (LE) type orbital distributions. To ensure a small ΔE_{ST} , a spatially separated LR-CT type orbital distribution is required. On the contrary, a large f needs efficient orbital overlap, that is a high ratio of LE type orbital distributions. It is therefore hard to make a compromise in molecular design and thus most previous works lean toward reducing ΔE_{ST} for a fast RISC without paying equal attentions to f value for an efficient radiative decay process. Our findings here provided an alternative orbital distribution, that is LR/SR-CT type, which is proved to facilitate a balanced small ΔE_{ST} and a large f for a short-lifetime delayed component. Our molecular design strategy should represent a conceptual advancement towards a new paradigm of TADF compounds, which possesses great potential to resolve the trade-off issue of a small ΔE_{ST} and a large f faced by solely LR-CT or LR-CT/LE type TADF emitters.

Moreover, we have also demonstrated a dual acceptor structure can further enlarge the f value without influencing ΔE_{ST} , which provided a screen that one can construct TADF emitters with equivalent multiple acceptors (or donors) to generate equivalent multiple CT channels to multiply the f value without influencing the small ΔE_{ST} . Interestingly, during the revision of our manuscript, we noticed that Prof. Kwon using a BO acceptor with two donors also benefits a faster k_{RISC} (*Nat. Commun.* **2023**, *14*, 419). But still, only LR-CT type orbital

transition was observed for their emitters (**Supplementary Table 8**).

Fig. The design principle for TADF emitters with different electronic excitations.

Besides, though the relatively broad emission spectra, conventional TADF molecules are still highly desired. On one hand, broad emission spectra are highly desired for white OLEDs as solid-state lighting applications. On the other hand, TADF molecules can also be utilized as sensitizers for fluorescent emitters, known as hyperfluorescence (HF) or TADF-sensitized fluorescence (TSF). Particularly, the combination of a TADF sensitizer and a MR final emitter has been a common strategy in literature, which resolves not only the issue of broad spectra of TADF molecules, but also the drawback of inefficient RISC process of MR compounds. Actually, based on such strategy, narrowband emission, high efficiency and long operation lifetime can be anticipated for OLEDs, which have been widely studied. A balanced k_f and k_{RISC} are highly desired for TADF molecules as not only emitters and also sensitizers. As an ideal sensitizer, a large k_{RISC} should benefit triplet capture and a large k_f favor to enhance the desired Förster energy transfer.

In the revised manuscript, we further constructed TSF devices using 1TICz, 1BOICz and 2BOICz as sensitizers for a previously reported yellow MR-TADF emitter, DBN-ICz. More details are described below in Comment (2), the TSF-OLEDs using 1BOICz and 2BOICz realized not only higher maximum EQEs but also smaller efficiency roll-off. The discrepancy can be attributed to the more balanced k_f and k_{RISC} of 1BOICz and 2BOICz, which not only facilitates an efficient T_1 up-conversion process but also favors fast energy transfer, thus quacking the exciton radiative consumption process to reduce the exciton accumulation under high current density. This finally leads to high device performance with low efficiency roll-off. The state-of-the-art performances also validated the superiority of molecule design strategy in developing not only efficient TADF emitters but also highly performed TADF sensitizer for MR emitters. Based on those discussions, we believed that our manuscript is sufficient enough and hope you can reconsider it.

2) Regarding the fast RICS and low-efficiency roll-off, the D-A or D-A-D typed TADF emitters have already been introduced. The TADF-OLEDs with emitters showed highly efficient performance of over 38% (EQE) with extremely low-efficiency roll-off, a high horizontal dipole ratio of over 90%, and a fast RISC of over 10^6 /s. However, the D-A or D-A-D typed TADF emitters use CT characteristics, which induce broadened EL emission similar to the D-MR-A emitter, resulting in low color purity.

Response: Thanks for your comments.

It is true that with the conceptual advancement in molecular design, some D-A or D-A-D/A-D-A typed TADF emitters have already realized high device efficiencies. However, most

TADF emitters still suffer from long-lifetime delayed components and thus poor device efficiency and stability under practical high luminance. To achieve a short-lifetime delayed component, an ideal TADF emitter requires not only a fast RISC but also a fast radiative decay. But still, most TADF emitters suffer from the mutual exclusion of a small ΔE_{ST} for a fast RISC and a large oscillator f for efficient radiation. From the point of view of orbital distributions, which essentially determined the performances of TADF emitters, previously D-A or D-A-D/A-D-A typed TADF emitters always inherited the LR-CT or LR-CT/LE type orbital distributions. A small ΔE_{ST} requires spatial orbital separation, that is a high ratio of LR-CT type orbital distributions. On the contrary, a large f needs efficient orbital overlap, that is a high ratio of LE type orbital distributions. It is therefore hard to make a compromise and thus most works lean toward reducing ΔE_{ST} for a fast RISC without paying equal attentions to f value for the efficient radiative decay process. Our findings here provided an alternative orbital distribution, that is LR/SR-CT type, which facilitates a balanced small ΔE_{ST} and a large f and may finally lead to a short-lifetime delayed component. Our molecular design strategy should represent a conceptual advancement toward a new paradigm of TADF compounds with a balanced small ΔE_{ST} and large f and finally a short-lifetime delayed component.

Though the relatively broad emission spectra, conventional TADF molecules are still highly desired. On one hand, broad emission spectra are highly desired for white OLEDs as solid-state lighting applications. On the other hand, TADF molecules can also be utilized as sensitizers for fluorescent emitters, known as hyperfluorescence (HF) or known as TADF-sensitized fluorescence (TSF). Particularly, the combination of a TADF sensitizer and a MR final emitter has been a common strategy in literature, which resolves not only the issue of broad spectra of TADF molecules, but also the drawback of inefficient RISC process of MR compounds. Actually, based on such strategy, narrowband emission, high efficiency and long operation lifetime can be anticipated for OLEDs, which have been widely studied.

A balanced k_r and k_{RISC} are highly desired for TADF molecules as not only emitters and also sensitizers. As an ideal sensitizer, a large k_{RISC} should benefit triplet harvesting and a large k_r favors to enhance the desired Förster energy transfer. We therefore envisioned that besides as emitters, 1BOICz and 2BOICz should also perform well as sensitizer for MR emitters. To verify this, TSF devices were fabricated using 1BOICz and 2BOICz as sensitizers, and a previously reported yellow MR-TADF emitter, DBN-ICz, was adopted as the narrowband emitter (*Adv. Mater.* **2022**, e2209396). For comparison, 1TICz was also adopted as sensitizer. All three TADF molecules provided efficient spectral overlapping between their emission spectra with the absorption spectrum of DBN-ICz emitter (**Supplementary Fig. 21**), affording large Förster resonance energy transfer (FRET) radius (R_0) of 3.4, 3.3, and 3.3 nm for 1TICz, 1BOICz, and 2BOICz based systems, respectively, which imply the efficient energy transfer in these sensitizing systems. Furthermore, all three films of mCPBC: 20 wt% sensitizer: 1 wt% DBN-ICz exhibited the same spectra peaking at 551 nm with a narrow FWHM of ~25 nm and the Φ_{PL} values of 94%, 97% and 98 % for 1TICz, 1BOICz and 2BOICz doped films, respectively.

The TSF-OLEDs were constructed with all the other functional layers that were the same as the TADF devices in the manuscript except the emitting layers of mCPBC: 20 wt% sensitizer: 1 wt% DBN-ICz. The detailed device configuration is depicted in **Supplementary Fig. 22** as well as the corresponding plot of current density (J), and luminance (L) versus

voltage (V), and power efficiency (PE), and current efficiency (CE) versus luminance. All three TSF devices displayed sharp spectra with peak at 551 nm, small FWHMs of 25 nm and CIE coordinates of (0.37, 0.61) (**Fig. 5**), representing pure orange emissions. Impressively high maximum external quantum efficiencies of 37.4% and 38.4% were obtained for the 1BOICz and 2BOICz based TSF devices, which is significantly higher than the one based on 1TICz (33.1%) as sensitizer. Moreover, the TSF devices based on 1BOICz and 2BOICz showed greatly alleviated efficiency roll-off, with EQE remaining 35.2% and 36.1% at 1,000 cd m^{-2} , respectively, being about 94% of their maximum values. On the contrary, an EQE of 27.0% was observed for 1TICz based device, being 81% of its maximum. The relatively higher efficiencies with smaller efficiency roll-off of devices based on 1BOICz and 2BOICz should be attributed to their more balanced k_r and k_{RISC} than those of 1TICz. The EL decay curves of those three TSF devices were also measured at 1,000 cd m^{-2} , showing obviously shorter delayed lifetimes of devices using 1BOICz and 2BOICz, which is similar to the results obtained from devices that used them directly as TADF emitters. This discrepancy can be attributed to the more balanced k_r and k_{RISC} of 1BOICz and 2BOICz, which not only facilitates an efficient T_1 up-conversion process but also favors fast energy transfer, thus quickening the exciton radiative consumption process to reduce the exciton accumulation under high current density. This finally leads to high device efficiency with low roll-off. The state-of-the-art performances validated the superiority of 1BOICz and 2BOICz as sensitizers and further evidenced the effectiveness of our molecular design.

Supplementary Fig. 21 (a) The chemical structure of narrowband yellow MR emitter DBN-ICz. (b, c, and d) UV-Vis absorption of DBN-ICz in toluene with a concentration of $10^{-5} \text{ mol L}^{-1}$, the PL spectra of 20 wt% 1TICz (or 1BOICz, 2BOICz) and 20 wt% 1TICz (or 1BOICz, 2BOICz): 1 wt% DBN-ICz doped in mCPBC.

Supplementary Fig. 22 (a) Device architecture for TSF-OLEDs based 1TICz, 1BOICz and 2BOICz. (b) Luminance-voltage-current density characteristics. (c) The current efficiency (CE) and (d) power efficiency (PE) versus luminance. (e) EL transient spectra for the TSF-OLEDs. (f-h) Angle-dependent EL intensities of the TSF-OLEDs with 2BOICz, 1BOICz and 1TICz, compared to the Lambertian distribution, the correction coefficient is calculated to be 0.980, 0.972, 0.970, respectively.

Fig. 5 TSF-OLED device performances. a. Normalized EL spectra under $1,000 \text{ cd m}^{-2}$. **b.** EQE versus luminance characteristics, and the narrowband yellow MR emitter DBN-ICz.

3) This reviewer agreed that the optoelectrical performance of OLEDs is impressive, however, could not find a novelty in the D-MR-A emitter compared to the D-A-typed TADF emitters in the previous literature. My concern is the D-MR-A emitter seems not to demonstrate the MR characteristics when connected to D and A molecules.

Response: Thanks for your valuable comments.

We are sorry for the misunderstanding induced by the “D-MR-A” abbreviation. In the revised manuscript, the description of the emitter has been changed to “TADF emitters with hybrid LR/SR-CT excitations”, namely TADF emitter with hybridized long-range (LR) and short-range (SR) charge transfer (CT) transitions. Furthermore, to better understand and strengthen the merits of this manuscript, we have changed the title to ‘*High-efficiency and stable short-delayed fluorescence emitters with hybrid long- and short-range charge-transfer excitations*’. We have been provided extensive revisions including in the Abstract and Introduction section to discuss the design principle for TADF emitters with different electronic excitations.

Previously reported D-A type TADF emitters always bear solely LR-CT or hybrid LR-CT/localized excited (LE) excitations, which face a mutual exclusion of a small singlet-triplet energy gap (ΔE_{ST}) and a large oscillator strength (f). This trade-off has been widely recognized as the main reason for a long-lifetime delayed component and will induce a significant efficiency roll-off and poor operation stability under high luminance. The aim of our work here is to propose a new paradigm of TADF emitters with hybrid LR/SR-CT excitations to break the above trade-off to balance a small ΔE_{ST} and a large f for a short-delayed component. As commented by Reviewer 1, our work here provided “the novel molecular design and outstanding device efficiency”.

Though their relatively wide spectrum, D-A type TADF emitters are still highly desired as their potential as not only emitters for lighting applications but also sensitizers for fluorescence emitters, known as hyper-fluorescence or TADF-sensitizer fluorescence (TSF). Particularly, the combination of a TADF sensitizer and a multi-resonance (MR) emitter has been a common strategy to avoid the inefficient RISC issue of MR emitter for high efficiency, narrowband emission and long device lifetime (*Nat. Photonics* **2021**, *15*, 203; *Nat. Photonics* **2021**, *15*, 208). An ideal TADF molecule as either emitter or sensitizer should possess a short-lifetime delayed component for a fast and efficient exciton consumption to guarantee good

device performances under practical high luminance. To achieve a short-delayed lifetime, a small ΔE_{ST} for an efficient RISC process and a large f value for efficient radiative decay should be satisfied simultaneously. However, in theory, a small ΔE_{ST} requires a spatially separated distribution of HOMO and LUMO while a large f requires an efficient overlap of HOMO and LUMO, which are principally conflicting with each other. The orbital distribution therefore determines the performances of molecules. Commonly, a long-range charge transfer (LR-CT) type orbital distribution means a large separation of HOMO and LUMO distributions, thus favoring a small ΔE_{ST} but also a small f value. The localized excited (LE) type orbital distribution, on the contrary, means an efficient HOMO-LUMO overlap, thus benefiting a large f value but also a large ΔE_{ST} . Nearly all D-A type TADF compounds in previous works can be divided into two categories as illustrated in the figure below, one is LR-CT type and another one is LR-CT/LE type. Considering that the LR-CT type would always favor a small ΔE_{ST} but also a small f , researchers developed TADF emitters with hybrid LR-CT and LE (LR-CT/LE) excitations, aiming to achieve a balance by controlling the ratio of LR-CT and LE orbitals. However, it is rather difficult and though some efficient TADF emitters have been developed, the progress is still tardy as most TADF emitters still suffer from a long-lifetime delayed component.

Recently, a new TADF emitter, that is the multi-resonance emitter, has emerged, bearing a new type of short-range CT (SR-CT) orbital distribution. This SR-CT excitation facilitates a similar or even larger f value and also a relatively smaller ΔE_{ST} compared with the LE type, but the ΔE_{ST} is not small enough to trigger an efficient RISC process compared with the LR-CT type. Besides, according to the Marcus-Levich-Jortner theory, the small Huang-Rhys factors of those molecules also render them rather slow RISC rates (k_{RISCs}) in the range of 10^4 s⁻¹ and even milli-second-scale τ_D (*Nat. Commun.* **2022**, *13*, 4876).

Inspired by the LR-CT/LE type emitters and considering the characters of those excitation types, we envisioned that hybridizing an auxiliary SR-CT with the main LR-CT excitation may not only avoid their disadvantages but also combine their complementary advantages to afford a new paradigm of TADF molecules simultaneously with a small ΔE_{ST} for a fast RISC process and a large f for an efficient radiative decay. With this in mind, we demonstrated a strategic implementation of TADF emitters with hybrid LR/SR-CT excitations by incorporating MR-acceptor groups on to a sterically uncrowded donor segment. This architecture, on one hand, created a hybrid electronic excitation with a main donor-to-acceptor LR-CT and an auxiliary bridge-phenyl SR-CT characteristic, balancing a small ΔE_{ST} and a large f . On the other hand, the adoption of equivalent dual multi-resonance acceptors was unveiled to double f without influencing ΔE_{ST} and also enlarged the horizontal dipole ratio for enhanced light extraction. The two proof-of-the-concept molecules successfully realized short-lifetime delayed components and unprecedented high EQEs with low roll-off and extended device lifetime. Our achievement here shatters the stereotypical physicochemical views of TADF emitters limited by the mutual exclusion of a small ΔE_{ST} and a large f , potentially revolutionizing the molecular design principle and unlocking the full potential of TADF OLEDs.

Previously D-A type TADF emitters beard only LR-CT or LR-CT/LE orbital distributions, difficult to balance a small ΔE_{ST} and a large f . And this is the first time that the concept of LR/SR-CT type TADF molecular was proposed, which should represent a conceptual advancement toward a new paradigm of TADF compounds with balanced small

ΔE_{ST} and large f . From the perspective of molecule structures, the LR/SR-CT molecular is also different with previous efficient TADF emitters. To guarantee a small ΔE_{ST} , a highly twisted structure is usually required for previously reported compounds. On the contrary, to satisfy LR/SR-CT orbital distribution, an uncrowded structure with a small dihedral angle is necessary. The relatively planar structure should favor not only a preferential horizontal oriented emitting dipole moment but also intrinsic molecular stability. From these points, we believe that a new paradigm of TADF molecules can be developed based on our concept, which is clearly different with the present molecules.

To make our idea clear, we chose three molecules for comparison as illustrated in the figure below. The 1TICz showed a moderate dihedral angle between the donor and acceptor due to its uncrowded structures and thus both HOMO and LUMO extended to the attached phenyl rings, showing LR-CT/LE type orbital distribution. Consequently, a large f value of 0.2361 was obtained, but also a large ΔE_{ST} of 0.1934 eV.

For TDBA-DI, owing to the large steric hindrance of the donor, a highly twisted structure with a large dihedral angle was obtained. A nearly complete separation of HOMO and LUMO was observed, indicating the main LR-CT type orbital distributions. A small ΔE_{ST} of 0.1934 eV was obtained, but also a small f value of 0.0309.

For 1BOICz, the uncrowded structure also renders a moderate dihedral angle and both HOMO and LUMO distributions were extended to the attached phenyl ring. Interestingly, unlike the situation in 1TICz, the HOMO and LUMO distribution on the attached phenyl ring was separated on single atoms, thus forming LR/SR-CT type orbital distribution. Consequently, a small ΔE_{ST} of 0.1054 eV and a large f value of 0.1426 was obtained, realizing a balance situation.

As 1TICz and TDBA-DI represent the most classical D-A type TADF emitters, it is clear that our molecular design showed a different concept regarding the orbital distributions, which should facilitate a new paradigm of TADF compounds to break the trade-off between a small ΔE_{ST} and a large f . Based on the above discussions, we believe that our findings are novel.

Figure. The design principle for conventional TADF emitters (LR-CT or LR-CT/LE type orbital distributions), and this work proposed hybrid LR-CT and SR-CT (LR/SR-CT) orbital distribution. The optimized structures of typical TADF emitter (TDDBA-DI) and the two compounds investigated in this work. The calculated HOMO-LUMO orbital distribution, related values of the dihedral angle (θ), singlet-triplet energy gap (ΔE_{ST}), and oscillator strength (f).

REVIEWER COMMENTS

Reviewer #1 (Remarks to the Author):

See Attachment.

The authors have provided more reasonable data in this revised manuscript. However, the authors rechecked important data to support the high EQE_{max} of 41.2%. There are still some concerns need to be resolved before the acceptance.

1. The authors have conducted 40 devices of 2BOICz with the EQE_{max} of 39.23 ±1.62%, while the authors chose EQE_{max} of 41.2% as the representative data. Is it reasonable? Moreover, the EQE_{max} of 2BOICz is 40.4% with Lambertian correction; thus, authors should only use the calibrated data in the manuscript. In addition, the authors mentioned the lower refractive index of the hole-transporting layer (TAPC compared to NPB) leads to the higher efficiency of the EL devices. However, the authors didn't provide a refractive index of the emitting layer (2BOICz). In addition, the authors should provide more robust evidence to support this very high efficiency, such as EQE_{max} simulation [DOI: 10.1002/adfm.201300104].
2. All the PLQYs of the three emitters are nearly 100%, and the delayed lifetimes of 1TICz, 1BOICz, and 2BOICz are 9889 ns, 968 ns, and 884 ns, respectively. Despite the significant difference in lifetime between 1TICz and other emitters, the fractional prompt/delayed fluorescence quantum yields (Φ_P/Φ_D) of 1TICz (93%/7%), 1BOICz (92%/8%) and 2BOICz (95%/5%) are similar, and Φ_P values of three emitters are so high compared to the references. Please explain Φ_P/Φ_D of these donor-acceptor emitters and list similar results of other work. We listed related data of the references from the Supplementary Table 8.

TADF materials	Refs	Φ_{PL}	Φ_P	Φ_D	τ_P (ns)	τ_D (μ s)
 TDBA-Ac	Nat. Photonics. 2019, 13, 540	0.93	0.48	0.45	35.1	1.82
 TDBA-DI	Nat. Photonics. 2019, 13, 540	0.99	0.51	0.48	15.2	1.79
 TMCz-BO	Nat. Commun. 2020, 11, 1765	0.98	0.66	0.32	38	0.75

 TMCz-3P	Nat. Commun. 2020, 11, 1765	0.76	0.65	0.11	29	14.5
 pMDBA-DI	Adv. Funct. Mater. 2021, 32, 2110356	0.978	0.468	0.51	45.6	1.80
 mMDBA-DI	Adv. Funct. Mater. 2021, 32, 2110356	0.973	0.532	0.447	47	2.90
 DBA-BFICz	Adv. Funct. Mater. 2021, 32, 2110356	0.919	0.80	0.119	5.79	2.93
 PzTDBA	Adv. Mater. 2021, 33, 2007724	0.998	0.319	0.679	44.6	2.63
 PzDBA	Adv. Mater. 2021, 33, 2007724	0.681	0.423	0.258	42.3	1.82
 DBA-DmICz	Nat. Commun. 2023, 14, 419	0.95	0.70	0.25	37.3	1.94
 DBA-DTMCz	Nat. Commun. 2023, 14, 419	0.99	0.51	0.48	36	0.92

3. Authors should provide temperature-dependent transient PL decay curves of the

three emitters in doped films at temperatures ranging from 77 K to 300 K.

4. In Supplementary Fig. 8c, whose emission spectra is it from, 2BOICz-tBu or the donor ICz-Ph? Please check it carefully. Also, in the text of Supplementary Fig. 9, 2BOICz-tBu is misspelled as 2BOICz-t.
5. Author suddenly remeasured the transient PL spectrum of 1TICz in this version and changed the fitting value. The delayed lifetime is changed from 3222 to 9889 ns. What happened? Please provide the responsible reason.

Reviewer #2 (Remarks to the Author):

accept

Reviewer #3 (Remarks to the Author):

To develop high-performance TADF-emitters with 100% Triplet to singlet conversion efficiency and reduced efficiency roll-off, tuning the energy gap between singlet and triplet and the oscillator strength of the emitter is one of the main concerns in this research field. A small S-T energy gap and a large oscillator strength are desirable to meet the purpose, but these are still challenging. In this manuscript, the authors introduce new TADF emitters to overcome the issue based on the hybrid long and short-range CT excitation. After the first revision, these concepts are well-addressed and more clearly organized in the manuscript. Therefore, I recommend it be accepted in the journal after addressing minor questions.

1. The authors realized new TADF emitters and demonstrated impressive device performance in the manuscript. The main point of the emitter is the D-A units linked with the MR core in the center to control the range of CT properties. However, the previously reported D-A-typed TADF emitter also demonstrated a high reverse ISC rate exceeding 10^6 s^{-1} with high efficiency and low-efficiency roll-off due to a short exciton lifetime of less than 1 usec using 3LE states. (For example, <https://doi.org/10.1038/s41566-020-0667-0>, <https://doi.org/10.1126/sciadv.abe5769>, <https://doi.org/10.1016/j.cej.2021.130224>). Therefore, it would be great to explain the benefit of the new emitter with MR core as superior to the conventional D-A typed emitters.
2. When looking at the spectra of the emitter with broad emission, CT has a primary role in the emitters instead of the MR core. Are there other candidates controlling long- and short-range CT characteristics instead of the MR core? Or MR core has a specialty to realize the character?
3. The manuscript indicates that managing the local-excited states is important. Recently, an MR-core-based TADF emitter to control the local T level and enhance the RISC rate was published so that it can be citable in this manuscript. (doi.org/10.1002/adma.202207416)

Response to reviewer's comments

Dear reviewers,

We thank all the reviewers for the helpful comments and suggestions, which have helped us to greatly improve our manuscript. We carefully considered all reviewers' comments and did our best to address these concerns. In the text below we provided a point-by-point response to all comments made by reviewers. The changes have been marked in the revised Manuscript and Supplementary Information.

REVIEWER COMMENTS

Reviewer #1 (Remarks to the Author):

See Attachment (It is copied below).

The authors have provided more reasonable data in this revised manuscript. However, the authors rechecked important data to support the high EQE_{max} of 41.2%. There are still some concerns need to be resolved before the acceptance.

1. The authors have conducted 40 devices of 2BOICz with the EQE_{max} of 39.23 ±1.62%, while the authors chose EQE_{max} of 41.2% as the representative data. Is it reasonable? Moreover, the EQE_{max} of 2BOICz is 40.4% with Lambertian correction; thus, authors should only use the calibrated data in the manuscript. In addition, the authors mentioned the lower refractive index of the hole-transporting layer (TAPC compared to NPB) leads to the higher efficiency of the EL devices. However, the authors didn't provide a refractive index of the emitting layer (2BOICz). In addition, the authors should provide more robust evidence to support this very high efficiency, such as EQE_{max} simulation [DOI: 10.1002/adfm.201300104].

Response: Thank you for your valuable comments.

(1) According to your suggestion, we have changed our expressions in the manuscript and only used the calibrated data. On the other hand, regarding the representative value and the statistic average one, we have noticed similar expressions in previous works (*Nature* **2021**, 599, 594-598; *Nature* **2022**, 611, 688-694). Therefore, we think it is reasonable to choose EQE_{max} of 40.4% as the representative data.

(2) The refractive index of the emitting layer was measured by a spectroscopic ellipsometer and is provided in **Supplementary Fig. 21**. A refractive index value of 1.75 at a wavelength of 528 nm was obtained.

(3) The optical simulation of the OLED device was performed by commercial software OFSS 1.0 (Wuhan Yuwei Optical Software Co., Ltd.). The input parameters include refractive index value, extinction coefficient, the thickness of each layer (all measured by ellipsometry), as well as photoluminescence (PL) spectrum, PLQY, and the ratio of horizontal emitting dipole orientation ($\Theta_{//}$) of the emitting layer (**Supplementary Fig. 21**). Assuming the recombination

zone is located in the middle of the emitting layer, the charge balance factor and singlet-triplet factor are set as unit, the simulation result of 2BOICz based TADF device exhibits a maximum EQE of 40.9%, matching well with the experimental result, indicating that the device efficiency was not overestimated.

Revised manuscript:

“The EQE versus luminance plots of those devices were illustrated in Fig. 3c and a significantly improved EQE_{max} up to 34.6% was obtained for 1BOICz, much higher than that of 1TICz (26.1%). A greatly alleviated efficiency roll-off was also noted for 1BOICz-based device, with EQE values of 33.8% and 31.7% at high luminance of 100 cd m⁻² and 1,000 cd m⁻². On the contrary, the EQE of 1TICz-based device sharply decreased to 24.9% and 16.7% at 100 cd m⁻² and 1,000 cd m⁻². The differences in efficiency roll-off behaviors were believed to be originated from the TADF emissive dynamics of the two emitters under electrical excitation, which will be depicted later by EL decay curves. Notably, 2BOICz-based device showed an exceptionally high EQE_{max} of 40.4%, calibrated using the angle-dependent EL distribution (Supplementary Fig. 18), which was maintained at 40.3% and 35.9% at 100 cd m⁻² and 1,000 cd m⁻², respectively. A maximum power efficiency (PE_{max}) of 122.4 lm W⁻¹ was also observed for 2BOICz as illustrated in Fig. 3d, obviously outperforming those of 1BOICz (95.5 lm W⁻¹) and 1TICz (71.7 lm W⁻¹), respectively. We have also measured the angle-dependent EL intensities of the devices based on 1TICz, 1BOICz and 2BOICz, and nearly Lambertian profiles pattern with the Lambertian coefficients of 0.97, 0.98 and 0.98 were recorded (Supplementary Fig. 18), respectively. Especially, the calibrated EQE_{max} value of 2BOICz device was still up to 40%, indicating that the high device efficiency was not overestimated. To confirm the reproducibility of the peak EQE_{max} of the device based on 2BOICz, ten groups of devices (40 testing points) were fabricated in parallel. The statistics of those EQE_{max}s show that the device performance is in good reproducibility.”

“Therefore, the excellent TADF properties, a high $\Theta_{//}$ and an optimized device structure should concurrently account for the cutting-edge EQE_{max} of 2BOICz. This is in accordance with the theoretically predicted EQE efficiency performed by a commercial software OFSS 1.0^{38,39} (Wuhan Yuwei Optical Software Co., Ltd.) (Supplementary Fig. 21).”

Table 2 Summary of the EL performance of TADF-OLEDs and TSF-OLEDs devices.

Device	Compound	λ_{EL}^a [nm]	FWHM [nm]	V_{on}^b [V]	EQE _{max/100/1,000} ^f [%]	PE _{max/100/1,000} ^e [lm W ⁻¹]	CIE (x,y) ^g
TADF	1TICz	504	100	3.1	26.1/24.9/16.7	71.7/61.6/37.1	(0.247, 0.499)
	1BOICz	534	101	3.1	34.6/33.8/31.7	95.5/92.1/76.0	(0.381, 0.566)
	2BOICz	528	91	3.0	40.4/40.3/35.9	122.4/114.4/89.6	(0.383, 0.550)
TSF	1TICz	551	25	3.0	32.2/32.1/26.3	88.0/87.4/60.9	(0.371, 0.611)
	1BOICz	551	25	3.0	36.6/32.9/34.4	99.3/94.5/87.5	(0.372, 0.612)
	2BOICz	551	25	3.0	37.6/36.2/35.4	102.3/101.2/89.4	(0.372, 0.613)

^aEL peak wavelength; ^bTurn-on voltage (V_{on}); ^cMaximum luminescence (L); ^dMaximum current efficiency (CE), value at 100 and 1,000 cd cm⁻²; ^eMaximum power efficiency (PE), value at 100 and 1,000 cd cm⁻²; ^fMaximum external quantum efficiency (EQE), value at 100 and 1,000 cd cm⁻². Both PE and EQE values of TADF and TSF devices are calibrated by Lambertian correction; ^gCIE coordinates at 1,000 cd cm⁻².

Supplementary Fig. 21 (a) The refractive index values of the functional layer and emitting layer (mCPBC: 20 wt% 2BOICz). (b) Theoretically simulated TADF device EQE value with respect to PLQY and horizontal dipole ratio. Optical simulations are performed by commercial software OFSS 1.0 (Wuhan Yuwei Optical Software Co., Ltd.)

2. All the PLQYs of the three emitters are nearly 100%, and the delayed lifetimes of 1TICz, 1BOICz, and 2BOICz are 9889 ns, 968 ns, and 884 ns, respectively. Despite the significant difference in lifetime between 1TICz and other emitters, the fractional prompt/delayed fluorescence quantum yields (Φ_P/Φ_D) of 1TICz (93%/7%), 1BOICz (92%/8%) and 2BOICz (95%/5%) are similar, and Φ_P values of three emitters are so high compared to the references. Please explain Φ_P/Φ_D of these donor-acceptor emitters and list similar results of other work. We listed related data of the references from the Supplementary Table 8.

TADF materials	Refs	Φ_{PL}	Φ_P	Φ_D	τ_P (ns)	τ_D (μ s)
TDBA-Ac	Nat. Photonics. 2019, 13, 540	0.93	0.48	0.45	35.1	1.82
TDBA-DI	Nat. Photonics. 2019, 13, 540	0.99	0.51	0.48	15.2	1.79
TMCz-BO	Nat. Commun. 2020, 11, 1765	0.98	0.66	0.32	38	0.75

 TMCz-3P	Nat. Commun. 2020, 11, 1765	0.76	0.65	0.11	29	14.5
 pMDBA-DI	Adv. Funct. Mater. 2021, 32, 2110356	0.978	0.468	0.51	45.6	1.80
 mMDBA-DI	Adv. Funct. Mater. 2021, 32, 2110356	0.973	0.532	0.447	47	2.90
 DBA-BFICz	Adv. Funct. Mater. 2021, 32, 2110356	0.919	0.80	0.119	5.79	2.93
 PzTDBA	Adv. Mater. 2021, 33, 2007724	0.998	0.319	0.679	44.6	2.63
 PzDBA	Adv. Mater. 2021, 33, 2007724	0.681	0.423	0.258	42.3	1.82
 DBA-DmICz	Nat. Commun. 2023, 14, 419	0.95	0.70	0.25	37.3	1.94
 DBA-DTMCz	Nat. Commun. 2023, 14, 419	0.99	0.51	0.48	36	0.92

Response: Thank you for your valuable comments.

The high Φ_P is actually one of the advantages for our molecule design, originating from the large oscillator strength (f). Under photoexcitation, only singlets are generated, and some of which will radiatively decay to the ground state to give prompt component, while another portion will flip to triplets and later generate delayed component. Therefore, a large f and thus a faster radiative decay rate (k_r) than ISC would lead to a large Φ_P . This is one of the main goals for TADF molecule design and would greatly reduce the exciton cycling between S_1 and T_1 states to guarantee a short delayed lifetime. For example, in 2BOICz, k_r is about 19 times larger than k_{ISC} , and almost all the up-converted excitons from T_1 to S_1 thereof would decay radiatively before go back to T_1 , thus only a small delayed component can be observed. (**Fig a**). This result is consistent with the purpose of this manuscript, establishing an ideal exciton dynamic model of $k_r \gg k_{ISC} \sim k_{RISC} > 10^6 \text{ s}^{-1}$.

In contrast, for conventional D-A type emitters listed in the table above, only a small oscillator strength (f) can be obtained owing to their small HOMO-LUMO overlap and thus moderate radiative decay rate but large k_{ISC} . Therefore, a large ratio of initial S_1 excitons will flip to the T_1 states and then generate delayed fluorescence. Even with a potentially large k_{RISC} , excitons in emitters with a slow k_r would repeat the $T_1 \leftrightarrow S_1$ cycles to provide a long-delayed lifetime (**Fig b**).

The difference in transient PL behaviors of the TADF emitters may originate from the different frontier orbital distributions, though all of which bear BO-type acceptors. Our emitters with hybrid long-range and short-range charge transfer (LR/SR-CT) transitions are advantageous as they combine a large f and a small ΔE_{ST} and finally a large Φ_P . Our emitters combined the advantages of both MR type and D-A type TADF emitters, realizing not only an efficient radiative decay but also a fast RISC process.

Fig. Photoluminescence processes of a) TADF emitters with hybrid LR/SR-CT excitations and b) conventional TADF emitters. k_r , k_{ISC} , k_{RISC} , k_{nr} , Φ_P represent fluorescence decay rate, intersystem crossing decay rate from S_1 to T_1 , reverse intersystem crossing decay rate from T_1 to S_1 , non-radiative decay rate from T_1 to S_0 , the efficiency of the prompt component of fluorescence.

3. Authors should provide temperature-dependent transient PL decay curves of the three emitters in doped films at temperatures ranging from 77 K to 300 K.

Response: Thank you for your valuable suggestions.

To investigate the nature of the delayed fluorescence behavior of three compounds. As shown in **Supplementary Fig. 8**, the temperature-dependent time-resolved photoluminescence

spectra of 1TICz, 1BOICz and 2BOICz doped into the mCPBC films (20 wt%) are recorded from 100 K to 300 K. With upon the increase of temperature, there is a clear increase in the emission intensity for all three compounds, which is characteristic of TADF emitters.

Supplemental Fig. 8 Temperature-dependent transient PL decay spectra of 1TICz (a), 1BOICz (b) and 2BOICz (c) doped into the mCPBC films (20 wt%).

Revised manuscript:

The PL decay curves and temperature-dependent decay spectra of the three doped films were also measured under an excitation wavelength of ~ 365 nm as illustrated in Fig. 2d-f and Supplementary Fig. 8, all exhibiting clear TADF behaviors with both prompt and delayed components.

4. In Supplementary Fig. 8c, whose emission spectra is it from, 2BOICz-tBu or the donor ICz-Ph? Please check it carefully. Also, in the text of Supplementary Fig. 9, 2BOICz-tBu is misspelled as 2BOICz-t.

Response: We are truly sorry for this mistake and the errors have been fixed in the revised Supplementary Information.

5. Author suddenly remeasured the transient PL spectrum of 1TICz in this version and changed the fitting value. The delayed lifetime is changed from 3222 to 9889 ns.

What happened? Please provide the responsible reason.

Response: Thank you for your comments.

In the original version of the manuscript, we measured the transient PL spectra of three emitters with the same time scale of the measurement window ($20 \mu\text{s}$) in order to show an obvious contrast for their excited state properties. However, we noticed that the time scale of the measurement window ($20 \mu\text{s}$) for the transient PL decay curve of 1TICz was not long enough. Therefore, a wider window with a time scale of $50 \mu\text{s}$ was adopted to accurately model it. And we have carefully re-conducted this measurement in the revised manuscript and reported a longer delayed lifetime of 9889 ns.

Reviewer #3 (Remarks to the Author):

To develop high-performance TADF-emitters with 100% Triplet to singlet conversion efficiency and reduced efficiency roll-off, tuning the energy gap between singlet and triplet and the oscillator strength of the emitter is one of the main concerns in this research field. A small S-T energy gap and a large oscillator strength are desirable to meet the purpose, but these are still challenging. In this manuscript, the authors introduce new TADF emitters to overcome the issue based on the hybrid long and short-range CT excitation. After the first revision, these concepts are well-addressed and more clearly organized in the manuscript. Therefore, I recommend it be accepted in the journal after addressing minor questions.

1. The authors realized new TADF emitters and demonstrated impressive device performance in the manuscript. The main point of the emitter is the D-A units linked with the MR core in the center to control the range of CT properties. However, the previously reported D-A-typed TADF emitter also demonstrated a high reverse ISC rate exceeding 10^6 s^{-1} with high efficiency and low-efficiency roll-off due to a short exciton lifetime of less than 1 usec using ^3LE states. (For example, <https://doi.org/10.1038/s41566-020-0667-0>, <https://doi.org/10.1126/sciadv.abe5769>, <https://doi.org/10.1016/j.cej.2021.130224>). Therefore, it would be great to explain the benefit of the new emitter with MR core as superior to the conventional D-A typed emitters.

Response: Thank you for your valuable comments.

The advantages of our molecular design tactic compared with the conventional D-A typed emitters can be summarized as: on one hand, our molecules can break the trade-off between a small ΔE_{ST} and a large oscillator strength (f); on the other hand, the relatively planar structures of our molecules naturally favor a horizontal orientation of emitting dipolar moments to enhance light extraction efficiency for a high EQE. Here are some more detailed explanations:

(1) It is true that some molecules also demonstrated short exciton lifetimes of less than $1\mu\text{s}$ in recent literature (cited as refs 29, 33 and 34 in our revised manuscript). Those papers mainly aimed to modulate ^3LE states of the TADF emitters so that they would be energetically close to ^3CT and ^1CT states for enhanced spin-orbital coupling. But most of those reported emitters still suffer from the mutual exclusion of a high k_{RISC} and a large k_{r} . For instance, Prof. Kaji et al. reported an emitter with a fast $k_{\text{RISC}} > 10^7 \text{ s}^{-1}$ by creating near-degenerate ^1CT , ^3CT and ^3LE states, which, however, only showed a slow k_{r} of 10^5 s^{-1} (*Nat. Photonics* **2020**, 14, 643-649). Besides, an even tougher but often ignored situation is the counter-effect from the intersystem crossing (ISC) process, which can be much faster than radiative decay and thus would increase the singlet-to-triplet spin-flip transition cycles to prolong the τ_{DS} of TADF emitters.

Our molecular design strategy here actually provides a totally different strategy to manipulate orbital distributions rather than modulating energetically-close ^3LE and CT excited states. By creating a hybrid long-range and short-range charge transfer (LR/SR-CT) transition, our molecular design can break the above trade-off of a large f and a small ΔE_{ST} , which is one of the main challenges for conventional TADF emitters bearing solely LR-CT character or a hybrid LR-CT/localized excited (LE) transition property. Therefore, in terms of the type of orbital distribution, our molecular design here represents a conceptual advancement. This LR/SR-CT transition can combine the advantages of LR-CT and SR-CT to

achieve a large f and a small ΔE_{ST} simultaneously, establishing an ideal exciton dynamic model of $k_r \gg k_{ISC} \sim k_{RISC} > 10^6 \text{ s}^{-1}$ for a short-delayed component of our molecules.

It is also believed that modulating the energetically-close 3LE and CT states would also work for our emitter with LR/SR-CT character, which may further improve the performances of such molecules. We also believed that introducing heavy atoms to enhance RISC may further shorten the delayed lifetime, which can be feasibly achieved by using sulfur-bridged boron (BS) MR acceptors. Such studies are now being conducted in our lab. We believed that TADF emitters with LR/SR-CT transition possess the great potential to unlock the full potential of TADF emitters.

(2) Furthermore, to enhance the EQE of OLEDs, a high horizontal emitting dipole orientation (EDO) ratio ($\Theta_{//}$) of emitters is highly desired to enhance light extraction efficiency. This required not only a preferential horizontal-orientated molecule but also a parallel EDO to molecular orientation. For conventional D-A type emitters, a highly twisted structure is usually required to achieve a small ΔE_{ST} . Though prolonging the length of molecules may enhance horizontal molecule orientations, the EDO of those molecules is not always parallel to molecular orientation and thus a relatively small $\Theta_{//}$, which finally would limit the device efficiency.

On the contrary, the emitters with LR/SR-CT transitions in our manuscript bear relatively planar structures to guarantee SR-CT. The planar structures have been proved to essentially favor an enhanced horizontal orientation of molecules. More importantly, the EDOs of those molecules are also tender to be parallel to molecular orientation. Therefore, a large $\Theta_{//}$ can be obtained for a high EQE. As a consequence, our emitters here achieved state-of-the-art EQE values of over 40% while the works in the above references only achieved maximum EQEs of below 30%.

Revised manuscript:

The following description has been added to the Introduction section: “It should be pointed out that some cutting-edge molecules in literature with mainly LR-CT transitions have also realized a short-delayed lifetime of $<1 \mu\text{s}$ by modulating the energy levels of LE triplet (3LE) and CT singlet (1CT) excited states.^{29,33,34} Our molecular design here is totally different with previous works as we focus on manipulating the orbital distribution. It is worth noting that previously reported strategies to enhance the RISC process should also work for our materials, which may further improve the performances of LR/ST-CT emitters.”

Reference:

29. Wada, Y., Nakagawa, H., Matsumoto, S., Wakisaka, Y. & Kaji, H. Organic light emitters exhibiting very fast reverse intersystem crossing. *Nat. Photonics* **14**, 643-649 (2020).

33. Lee, Y. H., Shin, Y.-S., Lee, T., Jung, J., Lee, J.-H. & Lee, M. H. Managing local triplet excited states of boron-based TADF emitters for fast spin-flip process: Toward highly efficient TADF-OLEDs with low efficiency roll-off. *Chem. Eng. J.* **423**, 130224 (2021).

34. Aizawa, N., Matsumoto, A. & Yasuda, T. Thermal equilibration between singlet and

triplet excited states in organic fluorophore for submicrosecond delayed fluorescence. *Sci. Adv.* **7**, eabe5769 (2021).

2. When looking at the spectra of the emitter with broad emission, CT has a primary role in the emitters instead of the MR core. Are there other candidates controlling long- and short-range CT characteristics instead of the MR core? Or MR core has a specialty to realize the character?

Response: Thank you for your valuable comments.

Considering the concept of our molecular design is to advantage of long-range (LR) and short-range (SR) charge transfer (CT) properties, we believe that an MR core is necessary to induce the SR-CT characteristics at present. But we can not rule out the possibility that other cores with SR-CT may be emerged in the future by deepening the understanding of the electronic properties of organic semiconductors.

Though depending on MR cores, our molecular design should be a general one as a great variety of MR cores have been developed with the advancement of both material science and synthetic methodology. Further investigations on different MR cores and molecule topology are currently being undertaken in our laboratory, which will generate a new paradigm of TADF emitters.

3. The manuscript indicates that managing the local-excited states is important. Recently, an MR-core-based TADF emitter to control the local T level and enhance the RISC rate was published so that it can be citable in this manuscript. (doi.org/10.1002/adma.202207416)

Response: Thank you for your valuable comments.

It is true that this paper did an excellent work to guarantee efficient exciton harvest for MR-TADF emitter by creating dense local triplet states in the vicinity of S_1 and T_1 energetically. We have cited this reference as 32 in the revised manuscript.

Reference:

32. Cheon, H. J., Woo, S. J., Baek, S. H., Lee, J. H. & Kim, Y. H. Dense Local Triplet States and Steric Shielding of a Multi-Resonance TADF Emitter Enable High-Performance Deep-Blue OLEDs. *Adv. Mater.* **34**, e2207416 (2022).

REVIEWERS' COMMENTS

Reviewer #1 (Remarks to the Author):

The authors have satisfactorily replied to almost all of my comments and concerns. I recommend that the manuscript is accepted for Nature Communications.

Reviewer #3 (Remarks to the Author):

The authors have well addressed the questions in the revised manuscript. Therefore, this reviewer recommends it be accepted in the journal.